# Noncanonical mechanism of voltage sensor coupling to pore revealed by tandem dimers of Shaker

João L. Carvalho-de-Souza [1,3] & Francisco Bezanilla[1,2]

In voltage-gated potassium channels (VGKC), voltage sensors (VSD) endow voltage-sensitivity to pore domains (PDs) through a not fully understood mechanism. Shaker-like VGKC show domain-swapped configuration: VSD of one subunit is covalently connected to its PD by the protein backbone (far connection) and non-covalently to the PD of the next subunit (near connection). VSD-to-PD coupling is not fully explained by far connection only, therefore an additional mechanistic component may be based on near connection. Using tandem dimers of Shaker channels we show functional data distinguishing VSD-to-PD far from near connections. Near connections influence both voltage-dependence of C-type inactivation at the selectivity filter and overall PD open probability. We speculate a conserved residue in S5 (S412 in Shaker), within van der Waals distance from next subunit S4 residues is key for the noncanonical VSD-to-PD coupling. Natural mutations of S412-homologous residues in brain and heart VGKC are related to neurological and cardiac diseases.

[1] Department of Biochemistry and Molecular Biology, The University of Chicago, Chicago, IL 60637, USA. [2] Centro Interdisciplinario de Neurociencia de Valparaíso, Facultad de Ciencias, Universidad de Valparaíso, Valparaíso, Chile. [3]Present address: Department of Anesthesiology, University of Arizona, Tucson, AZ 85724, USA. Correspondence and requests for materials should be addressed to F.B. (email: fbezanilla@uchicago.edu)

Voltage-gated potassium channels are often homotetramers, in which each subunit contains two structurally and functionally distinct domains: a voltage sensor domain (VSD; transmembrane segments S1–S4) and a pore domain (PD; segments S5–S6). A central $K^+$ selective conducting pathway is formed by the contributions of the four PDs. The VSDs, located peripherally to the central pore, endow the channel with its intrinsic voltage sensitivity by modulation of its open probability. The molecular basis of the VSD-to-PD functional coupling is not yet fully understood. The canonical coupling is thought to be done by an intracellular short α-helical segment, the S4–S5 linker[1–6]. More recently, studies have shown an additional non-canonical VSD-to-PD coupling mechanism that includes a role played by noncovalent interactions between amino acids side chains from VSD (S4) and PD (S5), henceforth VSD//PD interface[7–9]. The structure of the Shaker-like human voltage-dependent $K^+$ channel, $K_V1.2$[10], displays prominent domain swapping, consequently the VSD//PD interfaces are formed by VSDs and PDs from different subunits[5,10]. Other domain-swapped voltage-dependent ion channels have been identified in structural studies, including $K_V7.1$[6], $Ca_V1.1$[11], Nav1.4[12], and this dual-coupling mechanism is expected to be a general property of this channels superfamily (Fig. 1a).

In Shaker channels we have recently found an unexpected interaction between VSD and PD: in the presence of mutation L361R in S3–S4 extracellular linker, gating currents become sensitive to the W434F mutation, which is in a distant P-loop moiety of PDs. Mutant W434F is known to abrogate potassium conductance by enhancing C-type inactivation, allowing the easy recording of gating currents[13,14]. We found that the average $Q$–$V$ curve (gating charge vs voltage, where gating charge is the electric charge involved in channel gating) from channels containing both of these mutations ($VSD^{L361R}PD^{W434F}$) substantially differs from the average $Q$–$V$ curve from channels only mutated in their VSDs ($VSD^{L361R}PD^{wt}$). W434F induced shifts in the voltage–dependences of the $Q$–$V$ curves $V_0$ and $V_1$ (based on a three-state kinetic model) by +45 and +52 mV, respectively[15]. The abnormal $Q$–$V$ from $VSD^{L361R}PD^{W434F}$ was first noticed when compared with the average conductance–voltage ($G$–$V$) curve from channels containing L361R mutation but without W434F mutation in their PD ($VSD^{L361R}PD^{wt}$): normalized $Q$–$V$ curve from $VSD^{L361R}PD^{W434F}$ mutants crosses $G$–$V$ curve from $VSD^{L361R}PD^{wt}$. However, when the same mutant ($VSD^{L361R}PD^{wt}$) is used for both $G$–$V$ and $Q$–$V$ curves (the latter after $K^+$ depletion), no crossing is observed, suggesting that the W434F mutation, located in the P-loop of the PD can somehow functionally affect the dynamics of a mutant VSD. With this knowledge we speculated that allosteric PD-to-VSD interactions could also be mediated through a hypothetically functional VSD//PD interface suggested by previous studies, including ours[7–9], in addition to the direct S4–S5 linker connection between VSD and PD.

Here, using a tandem dimer approach we show evidence suggesting that, in addition to the S4–S5 linker, VSDs confer voltage dependence to PD via this unique putative VSD//PD interface. We hypothesize this interface is part of an important non-canonical coupling mechanism present in domain-swapped voltage-dependent ion channels. This information provides strong foundation for the study of VSD-to-PD coupling mechanisms in voltage-gated ion channels.

## Results

### Shaker dimerization as strategy to study VSD//PD interfaces.
The main goal of this study was the investigation of the functionality of the interface formed by a VSD and a PD from different

subunits in a domain-swapped voltage-dependent channel. We focused on the fast-inactivation-removed Shaker channel, henceforth just Shaker, in which we mutated its VSD and/or PD. Mutations in VSD were basically used to produce different voltage dependencies while a mutation in PD enhanced the C-type inactivation feature of the PD thus providing a readout of this domain. In summary, the putative functionality of the VSD//PD interface is measured comparing the shifts of the voltage dependence of the C-type inactivation process with the shifts in the voltage dependence of the VSD, previously studied by $Q$–$V$ curves.

Mutations in Shaker protein are represented fourfold in the channels. W434F mutations yield nonconductive channels, avoiding electric readout of the PD. However, when used twofold per channel, by expressing Shaker as dimers of dimers, W434F mutations do not abrogate $K^+$ conductance, consequently allowing an electric readout of the PD. The study of the gating currents produced by these channels is impractical (see Supplementary Note 1).

### Comparing Shaker dimers with Shaker.
We created tandem-duplicated Shaker cDNA and the expression of it is henceforth called Shaker dimers, to differentiate them from the homotetramer Shaker (Compare Fig. 1a with Fig. 1b). One at a time, we mutated, PD from protomer 1 or protomer 2 and the expression of these twofold W434F-contaning Shaker dimers were named W434F-Shaker dimers. Both types of channels express well in the membrane and their voltage dependences for activation ($V_{1/2}$) taken from conductance–voltage ($G$–$V$) curves were not significantly altered relative to Shaker (Fig. 1c–e and Supplementary Table 1). In W434F-Shaker dimers C-type inactivation shows reproducible differences depending whether $PD^{W434F}$ is placed in protomer 1 or in protomer 2. The kinetics of the inactivation process is described by two components: fast and slow. When $PD^{W434F}$ is in protomer 1, the fast component has a time constant of 0.4 s, but when placed in protomer 2 this time constant slows down to about 1 s (Fig. 1f). However, regardless the position of $PD^{W434F}$, the slow component of inactivation is essentially unaltered, ranging from 1.5 to 4 s.

The relative contribution of the fast component of inactivation (fast/(fast + slow)) in W434F-Shaker dimers also changes according to the position of $PD^{W434F}$. In Shaker dimers, the fast component contributes to 40% of the total inactivation process. When W434F mutation is present in protomer 1 this percentage increases to 85% and when the mutation is placed in protomer 2, it reaches 65% (Fig. 1g). We interpret this difference as caused by possible radial asymmetries in the channel produced by $PD^{W434F}$ with or without C-terminal-free S6. This point will be taken up in the "Discussion" section.

Voltage-dependent inactivation curves (Inact-$V$) were constructed from the degree of inactivation after 10 s periods at different voltages. After 10 s at depolarizations to −10 mV and beyond, Shaker dimers inactivate to 30% of their peak, similar to Shaker (Fig. 1h), while W434F-Shaker dimers inactivate to 15% of its peak under the same conditions. W434F-Shaker dimers inactivate at more negative voltages than Shaker dimers, as observed by the $V_{inact}$ taken from two-state model fittings (see "methods" section and Supplementary Table 1), indicating that some inactivation occurs without activation since their $G$–$V$ curves are unchanged.

### Evidence of a functionally active VSD//PD interface.
We studied the voltage dependence of the inactivation process in W434F-Shaker dimers when different $VSD^{mut}$ are present in the same VSD//PD interfaces. Changes in $V_{inact}$ were compared with changes in voltage dependences of related $VSD^{mut}$ ($V_0$ and $V_1$

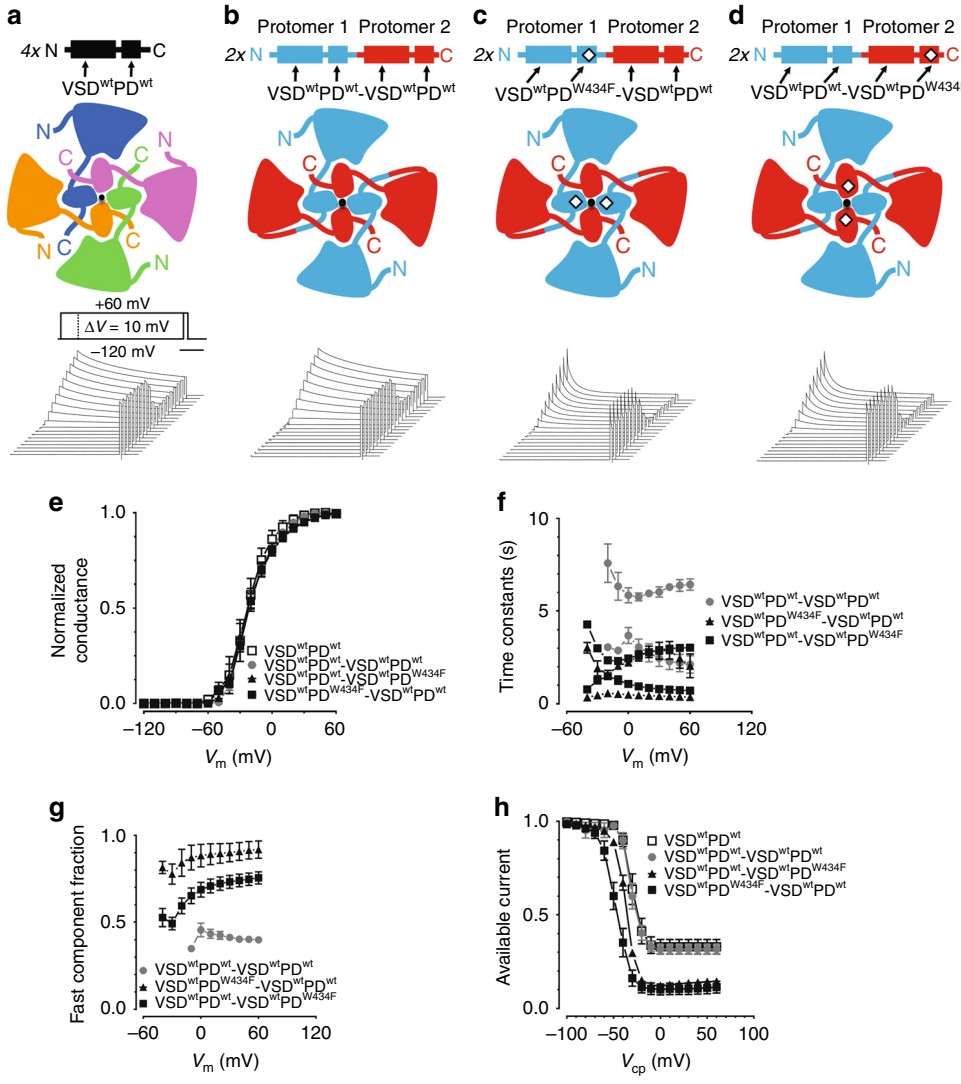

**Fig. 1** Biophysical properties of Shaker dimers when mutated in PDs. **a** Top: representation of Shaker protein showing wild-type voltage sensor (VSD$^{wt}$) and pore domains (PD$^{wt}$); Middle: Top view of Shaker channel in the membrane, showing each subunit (protein) of the tetramer represented by a different color, with N and C representing amino-terminus and carboxy-terminus, respectively. The black circle in the center represents K$^+$ ions within the conductive pathway. Bottom: Typical K$^+$ currents from Shaker channels expressed in Xenopus oocytes with the indicated voltage protocol. Horizontal bar means 2 s. **b** Top and middle: schematics showing dimerized Shaker, with protomer 1 in blue and protomer 2 in red. These channels do not contain W434F mutations and are called wt-PD Shaker dimers. Bottom: K$^+$ currents recorded from wt-PD Shaker dimers in similar condition as in **a**. **c**, **d** Top and middle: representation of Shaker dimers containing the mutation W434F (W434F-Shaker dimers) in protomer 1 in **c** and in protomer 2 in **d**. The mutant domains are marked by a diamond. Bottom: K$^+$ currents recorded with the respective channels in similar conditions as in **a**. In **e**–**h**, open squares are for Shaker, gray circles for wt-PD Shaker dimers, black triangles, and black squares for W434F-Shaker dimers with W434F mutation in protomers 1 and 2, respectively. **e** Normalized conductance–voltage (G–V) curves for the channels represented in **a**–**d**. **f** Time constants of the K$^+$ currents inactivation in the indicated channels during 10 s depolarizing voltages periods (same protocol shown in **a**, bottom, and same symbols convention as in **e**). Fast and slow time constants from two exponentials fit to the data (see "methods" section for details), are shown for all Shaker dimers as in **b**–**d**. **g** Fast component relative amplitude of the exponential components in the channels indicated. **h** K$^+$ currents 10 s inactivation curves (Inact-V) in the channels indicated. All data points are the average of 4–6 independent experiments. The vertical bars in **e**–**h** are the standard error of the mean. VSD$^{wt}$PD$^{wt}$: Shaker; VSD$^{wt}$PD$^{wt}$-VSD$^{wt}$PD$^{wt}$: wt-PD Shaker dimer; VSD$^{wt}$PD$^{W434F}$-VSD$^{wt}$PD$^{wt}$ and VSD$^{wt}$PD$^{wt}$-VSD$^{wt}$PD$^{W434F}$: W434F-Shaker dimers with W434F in protomer 1 and in protomer 2, respectively

from Q–V curves). If $V_{inact}$ relates to $V_0$ and/or $V_1$, this would be a functional demonstration of the VSD//PD interfaces as part of the VSD-to-PD coupling. To this end, we created dimers with many different types of VSD$^{mut}$, one at a time, as means to have VSD with different $V_0$ and $V_1$ for the observation of $V_{inact}$ from the PD.

We first introduced the well-studied ILT mutant[16] (mutations V369I, I372L, and S376T) in one VSD of W434F-Shaker dimers (VSD$^{ILT}$). In Shaker, ILT changes the VSD-to-PD coupling by

increasing the energetic barrier for the last of VSD during activation. This shifts the G–V curve by 100+ mV, but it does by shifting only the final 15% charge movement. We found in ILT mutations an excellent opportunity to separate Q–V curve in two parts during a series of depolarizing pulses from negative voltages: 85% initial movements, nonrelated to pore opening, and the final 15%, related to open probability.

In order to generate a channel that has VSD and PD mutations in the same VSD//PD interfaces, near to each other, VSD$^{ILT}$ and

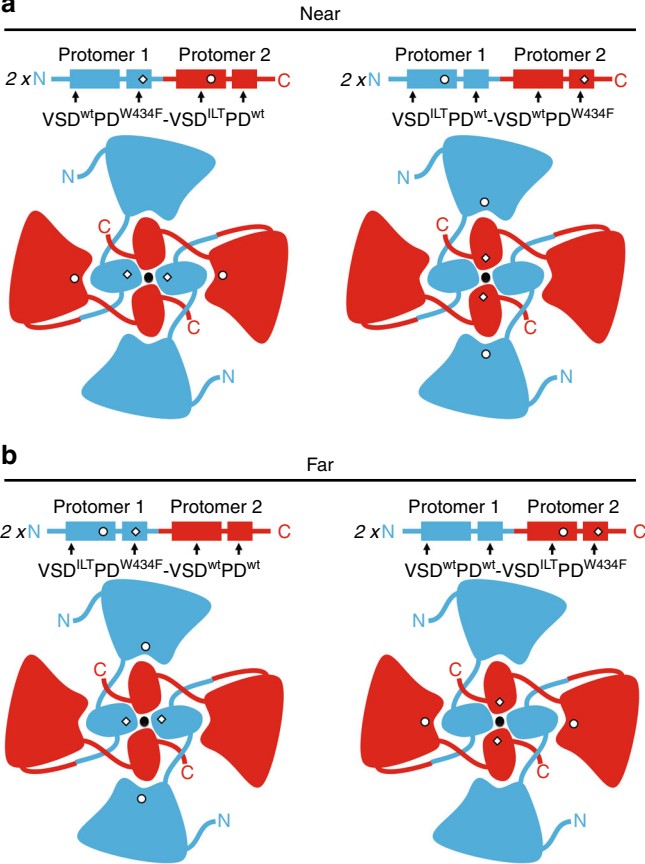

**Fig. 2** Schematic representation of near and far configuration Shaker dimers. **a** Near configuration dimers shown in two possible designs, both with $VSD^{ILT}$ and $PD^{W434F}$ in different protomers ($VSD^{wt}PD^{W434F}$-$VSD^{ILT}PD^{wt}$ or $VSD^{ILT}PD^{wt}$-$VSD^{wt}PD^{W434F}$). Protomer 1 is shown in blue and protomer 2 is shown in red in the top left and right panels with the mutant VSD indicated by circles and mutant PD indicated by diamonds. Note that mutant domains are in different protomers in both cases. **b** Far configuration dimers shown in their two possible designs, where $VSD^{ILT}$ and $PD^{W434F}$ are always in the same protomer ($VSD^{ILT}PD^{W434F}$-$VSD^{wt}PD^{wt}$ or $VSD^{wt}PD^{wt}$-$VSD^{ILT}PD^{W434F}$). Protomers are color coded as in **a**. Similarly, the mutant VSD are indicated by circles and mutant PD indicated by diamonds

$PD^{W434F}$ were introduced in different protomers. This strategy was followed in the two W434F-Shaker dimers backgrounds, with $PD^{W434F}$ in protomer 1 and protomer 2, thus testing also the possible asymmetry of the final channel: (i) $VSD^{ILT}$ in protomer 1 and $PD^{W434F}$ in protomer 2 ($VSD^{ILT}PD^{wt}$-$VSD^{wt}PD^{W434F}$) and (ii) $VSD^{ILT}$ in protomer 2 and $PD^{W434F}$ in protomer 1 ($VSD^{wt}PD^{W434F}$-$VSD^{ILT}PD^{wt}$). In both cases two VSD//PD interfaces are formed by mutants VSD and PD, and the other two VSD//PD interfaces are formed by wild-types VSD and PD. In these near configuration dimers, changes in inactivation (caused by $PD^{W434F}$ mutations) that follows the voltage dependence of $VSD^{ILT}$ are putatively dependent on the $VSD^{ILT}$//$PD^{W434F}$ interfaces. (Fig. 2a).

Also using the two W434F-Shaker dimers backgrounds, VSD and PD were mutated in the same protomer: (i) in protomer 1, $VSD^{ILT}PD^{W434F}$-$VSD^{wt}PD^{wt}$ or (ii) in protomer 2 $VSD^{wt}PD^{wt}$-$VSD^{ILT}PD^{W434F}$. The resultant channels contain the same mutations as the near configuration dimers, but since mutants VSD and PD are located in the same protomer, they are far apart from each other, never forming the same VSD//PD (Fig. 2b).

Robust $K^+$ currents were obtained from all four constructs (two in near and two in far configurations) (Fig. 3a–d).

Relative to $G$–$V$ curves from Shaker, $G$–$V$ curves from dimers containing $VSD^{ILT}$ in near configuration are more displaced in voltage than when in far configuration (compare orange with green symbols in Fig. 3e, f). While the effect of $VSD^{ILT}$ in Shaker (four $VSD^{ILT}$ per channel) is a +112.0 mV shift in $V_{1/2}$, in near configuration dimers $V_{1/2}$ is shifted by +77.3 or +85.3 mV, and in far configuration dimers $V_{1/2}$ are shifted by only +41.5 and +6.5 mV (Fig. 3i and Supplementary Table 2). The voltage dependence of the C-type inactivation is also differently affected in near versus far configurations. In near configuration, the inactivation curves are split and roughly follow the two components of the $VSD^{ILT}$ $Q$–$V$ curve (Fig. 3g, h). One component is essentially unchanged relative to the inactivation curve from the respective W434F-Shaker dimer (Fig. 3e, f). These results indicate that in the near configuration channels two types of inactivation processes take place: one that follows the open state and a second one, independent of the open state and that follows the movement of the $VSD^{ILT}$ (not correlated with PD conductance). In the far configuration, the inactivation curves are monotonic, very similar to the curves from the respective W434F-Shaker dimer (Fig. 3e, f) and mostly follow the $G$–$V$ curve of the same channel, suggesting the whole inactivation process is governed by the open channel.

**VSDs control activation and inactivation in near configuration channels.** To test the generality of the results presented above, we introduced other different $VSD^{mut}$ in W434F-Shaker dimers in near and far configurations. Earlier, we found single amino acid mutations in S3–S4 linker of VSD that can shift its voltage dependence over a broad voltage range[15]. Because the VSD moves from resting to active state in at least two distinguishable steps, the voltage dependence for each $VSD^{mut}$ is given by two voltages, $V_0$ and $V_1$. $V_0$ represents the midpoint of the first step that is not related to channel opening. $V_1$ is the midpoint of the second step that correlates with the open probability of the $K^+$ conductance. $VSD^{mut}$ voltage dependencies ($V_0$ and $V_1$) along with the voltage dependence of the conductance activation ($V_{1/2}$) are depicted in Fig. 4a. Here, we used two dimer configurations: $VSD^{wt}PD^{W434F}$-$VSD^{mut}PD^{wt}$ as near configuration and $VSD^{wt}PD^{wt}$-$VSD^{mut}PD^{W434F}$ as far configuration of W434F-Shaker dimers. This decision was based on the fact that the mutation in the VSD of protomer 2 gives more contrast when comparing the two cases of W434F-Shaker dimers, near versus far configurations.

Inactivation in W434F-Shaker dimers in near configuration is fundamentally different from far configuration. In the near configuration dimers, much more than in the far configuration ones, $V_{inact}$ are more related to voltage dependence of $Q$–$V$ curves (Fig. 4b, c). This is seen from broader voltages range of the inactivation curves for instance when $VSD^{ILT}$ is placed for near, compared with for far configuration. Inact-$V$ curves were analyzed by fitting a two-state model (noninactivated and inactivated), generating a voltage dependence parameter $V_{inact}$ for each curve (Fig. 4f and Supplementary Table 2). In most near configuration dimers the model was able to fit the data as it can be seen in the sum of squared residuals shown in Fig. 4f. In the cases of dimers containing $VSD^{361A}$ and $VSD^{ILT}$, mutant voltage sensors exhibiting split $Q$–$Vs$, the two-state model was unable to explain the inactivation curves splitting features shown in Fig. 4b. This split inactivation suggesting at least two inactivating processes taking place led us to use two independent and weighed two-state models to fit the data. This analysis generates two voltage-dependent parameters for the inactivation of these channels, $V_{1st}$ and $V_{2nd}$, resulting

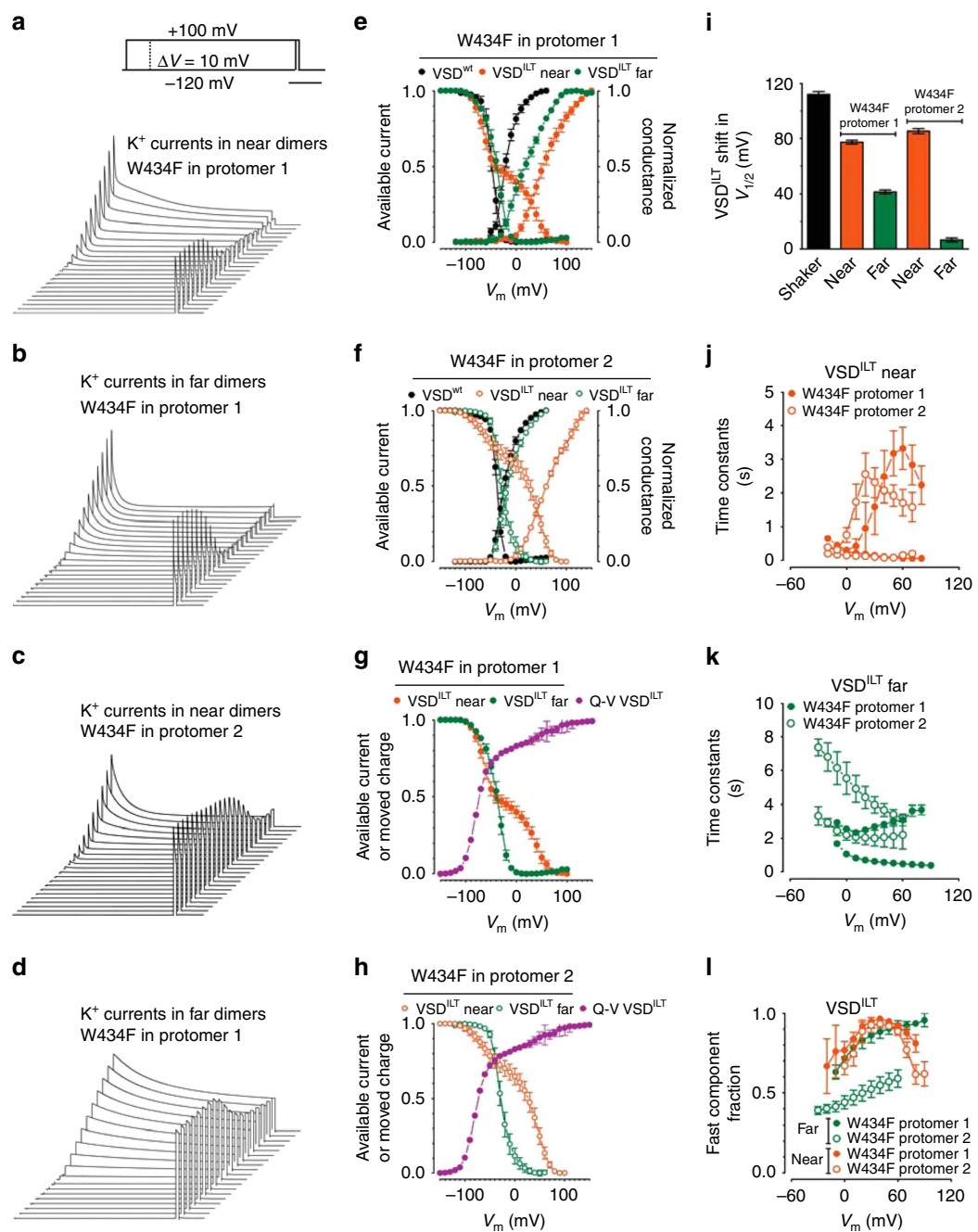

**Fig. 3** Functional characterization of all types of W434F-Shaker dimers bearing VSD[ILT]. K+ currents from dimers with PD[W434F] mutation in protomer 1 are shown in **a** and **b**, configured as near and far channels, respectively. Horizontal bar in **a**, top, means 2 s. In **c** and **d**, K+ currents are shown from dimers with PD[W434F] mutation in protomer 2, configured as near and far, respectively. All currents in **a–d** were activated by the same voltage protocol as shown in **a**, top panel. G–V and Inact-V curves are shown in **e** for near (orange) and far (green) dimers with W434F in protomer 1 and in **f** with W434F in protomer 2. Curves in black shown in **e** and **f** are from the corresponding control, PD[W434F] in protomer 1 and in protomer 2, respectively, and with no VSD mutated (see Fig. 1). Inact-V curves in W434F-Shaker dimers, near and far, with the PD[W434F] in protomer 1 (**g**) and in protomer 2 (**h**) are also shown together with the Q–V curve for VSD[ILT]. **i** The shift in G–V curves introduced by the presence of VSD[ILT] in Shaker (black bar) and in the four types of W434F-Shaker dimers, as labeled in the graph. **j** and **k** show K+ currents inactivation fast and slow time constants taken from two exponential fits (see "methods" section) from near and far configuration dimers, respectively. In both graphs filled and empty circles (orange or green) are for dimers with PD[W434F] in protomer 1 and in protomer 2, respectively. **l** Fraction of the fast inactivating component are shown for near and far dimers with the position of the PD[W434F] indicated by the labels in the graph. All data points are the average of 4–6 independent experiments. The vertical bars in **e–l** are the standard error of the mean

in a satisfactory sum of the squared residuals of the fit (Fig. 4f). A single two-state model was enough to fit the inactivation curves from the far configuration dimers with low residuals (Fig. 4g).

To look for possible correlations between VSD movement and inactivation, voltage dependences for inactivation ($V_{inact}$ or $V_{1st}$ and $V_{2nd}$), and for VSD ($V_0$ and $V_1$) were used to calculate shifts in voltage induced by the VSD[mut]. Therefore, shifts in $V_{inact}$ from

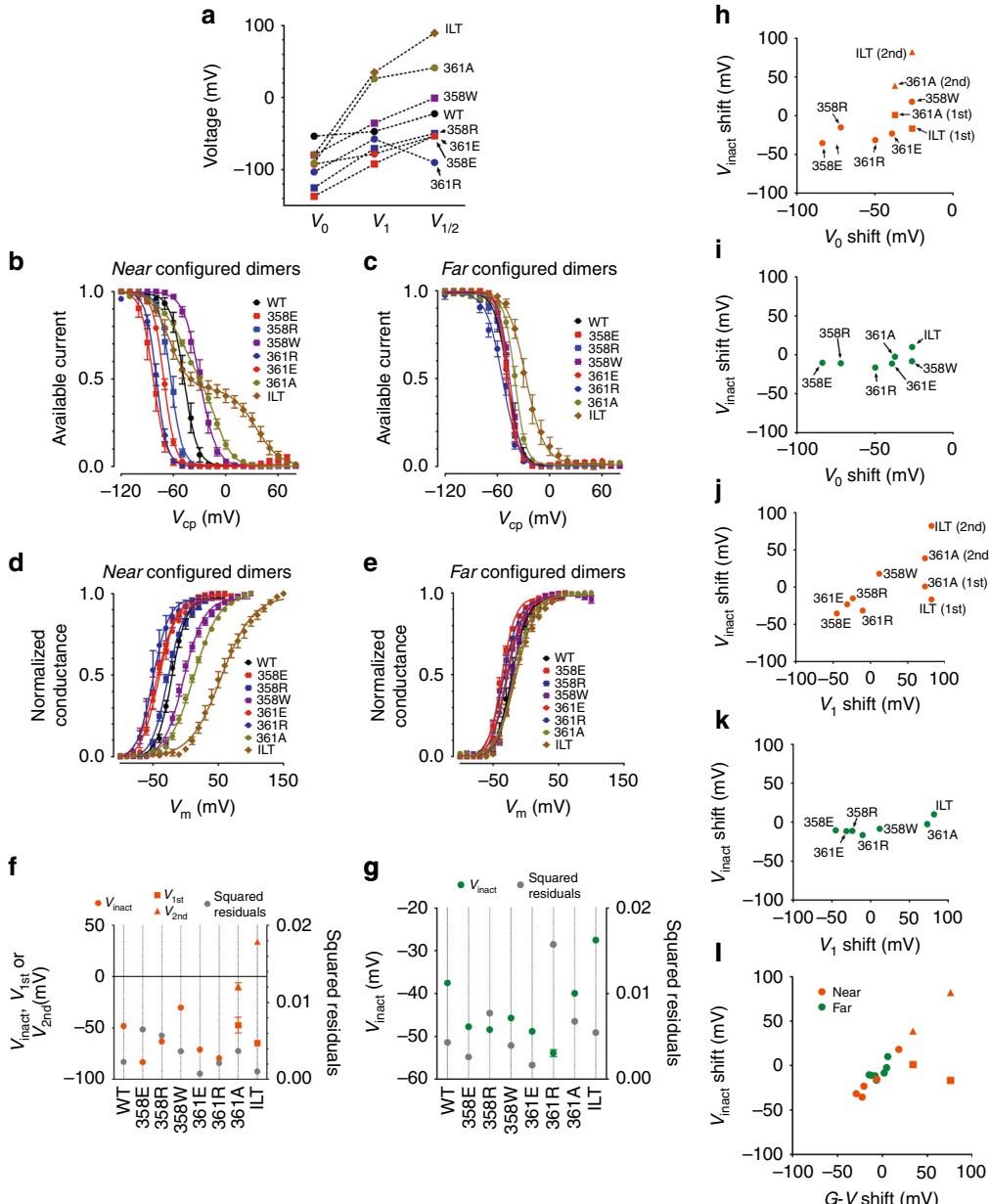

**Fig. 4** Study of the VSD$^{mut}$//PD$^{W434F}$ interfaces in W434F-Shaker dimers. **a** Voltage dependences of Shaker containing different VSD$^{mut}$. Plots show three voltage dependences of Shaker channels with different VSDs: $V_0$ and $V_1$ from $Q$–$V$ and $V_{1/2}$ from $G$–$V$ curves. **b**, **c** Inact-$V$ curves from W434F-Shaker dimers, near and far configuration, respectively and as indicated, for each VSD$^{mut}$ case. Mutations in the VSD are color/shape coded and are shown in the insets. **d**, **e** $G$–$V$ curves for the same channels, near and far, shown in **b** and **c** with the mutations indicated by same color/shape code as in **b** and **c**. All Inact-$V$ curves, except from near dimers containing VSD$^{361A}$ or VSD$^{ILT}$, were fitted by a two-state model: noninactivated and inactivated (see "methods" section for details), generating a voltage-dependent parameter $V_{inact}$ (Supplementary Tables 1 and 2). In near dimers containing VSD$^{361A}$ or VSD$^{ILT}$, the curves were fitted by two independent two-state models to generate two voltage-dependent parameters: $V_{1st}$ and $V_{2nd}$. **f**, **g** Plots of $V_{inact}$, $V_{1st}$, and $V_{2nd}$ from near and far different dimers as labeled, respectively. Squared residuals from the fittings that generated the voltage-dependent parameters $V_{inact}$, $V_{1st}$, and $V_{2nd}$ are also plotted and connected to them by the vertical dotted lines. Shifts in $V_{inact}$ (or in $V_{1st}$ and $V_{2nd}$) from near (orange circles for $V_{inact}$, squares for $V_{1st}$ and triangles for $V_{2nd}$, **h** and **j**) and far dimers (green circles, **i** and **k**) relative to the values in the respective Shaker dimers with VSD$^{wt}$ only plotted against the respective shifts in $V_0$ (**h** and **i**) and $V_1$ (**h** and **i**) from VSD$^{mut}$ taken from $Q$–$V$ curves in nonconductive Shaker. **l** The same shift values of $V_{inact}$, $V_{1st}$ and $V_{2nd}$ shown in **h**–**k** from near and far dimers were plotted against the shifts in $V_{1/2}$ in Shaker induced by the respective VSD$^{mut}$ (also relative to channels with VSD$^{wt}$ only). Labels at each data point were omitted for clarity. All data points are the average of 4–6 independent experiments. The vertical bars in **b**–**g** are the standard error of the mean. The error bars in plots **h**–**l** were omitted for clarity

near and far configuration dimers were plotted against the shifts in $V_0$ (Fig. 4h, i) and in $V_1$ (Fig. 4j, k) for same VSD$^{mut}$. Altogether, data show there is a mutual relationship or connection between the shifts in $V_0$ and $V_1$ with the shift in $V_{inact}$ (or in $V_{1st}$ and $V_{2nd}$) for the near configuration dimers

(orange symbols) but not much for far configuration dimers (green symbols).

$G$–$V$ curves from both near and far configuration dimers were also analyzed (Fig. 4d, e). VSD$^{mut}$-induced shifts in voltage dependences of $G$–$V$ curves from dimers were clearly larger in

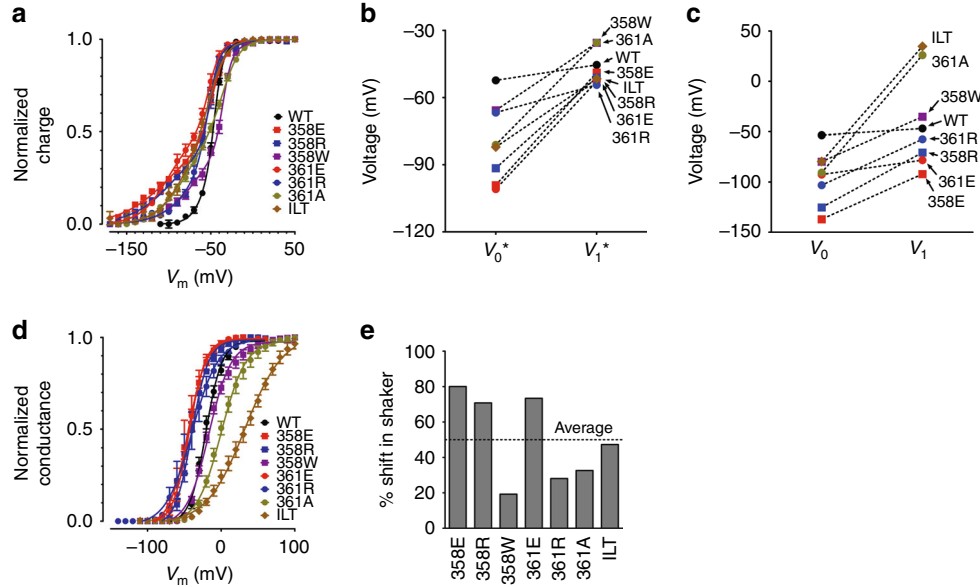

**Fig. 5** wt-PD Shaker dimers (no W434F mutation) and nonconductive Shaker dimers (all PD mutated with W434F) show independent VSD domains. **a** $Q$–$V$ curves are shown for all nonconductive dimers studied including dimers with only $PD^{W434F}$ and the ones possessing two $VSD^{mut}$, indicated in the inset in a color/shape code. The curves were fitted with two independent two-state models for the generation of two voltage dependences values, $V_0^*$ and $V_1^*$. These values are plotted in **b** and for comparison, $V_0$ and $V_1$ (from nonconductive Shaker) are plotted in **c**. **d** $G$–$V$ curves from wt-PD Shaker dimers possessing only $VSD^{wt}$ and the ones possessing two $VSD^{mut}$, similarly indicated in the inset in a color/shape code. Data were fitted with a two-state model and the values of the $G$–$V$ midpoints ($V_{1/2}^*$) were used to calculate the shifts produced by the mutations in the VSD as compared with $V_{1/2}^*$ from wt-PD Shaker dimers that only contains $VSD^{wt}$ (**e**). A horizontal dotted line shows the overall average value of the % shift and its value is 49.8%. All data points are the average of 4–6 independent experiments. The vertical bars in **a** and **d** are the standard error of the mean. The plots in **b** and **c** were omitted of error bars for clarity. Continuous lines over the data points represent the best fit of two independent two-state models for $Q$–$V$ curves in **a** and a two-state model for $G$–$V$ curves in **d**

near than in far configuration dimers. A two-state model successfully fits the activation curves generating the voltage-dependent parameter $V_{1/2}$ (Supplementary Table 2). The functional coupling between activation and inactivation, a hallmark of C-type inactivation, was also tested by plotting $V_{1/2}$ for each mutant, separated as near or far, against their respective $V_{inact}$ (or $V_{1st}$ and $V_{2nd}$) from their Inact-$V$ curves. These plots depict how the inactivation process is dependent upon activation, and they demonstrate that the inactivation process is, at least in part, dependent on the activation process as was previously established for C-type inactivation processes[17] (Fig. 4l).

**wt-PD and nonconductive Shaker dimers show independent VSD**. Finally, we made two set of Shaker dimers, one with C-type inactivation fully sped up by the presence of $PD^{W434F}$ in both protomers, yielding a channel with four W434F mutations (all $PD^{W434F}$), therefore nonconductive (nonconductive Shaker dimers) and another where no $PD^{W434F}$ mutation is present (all $PD^{wt}$) yielding channels with normal C-type inactivation therefore (wt-PD Shaker dimers). The nonconducting Shaker dimers represented a strategy to study gating currents in channels with two out of the four VSD mutated in order to produce significant shifts in their voltage dependence. The wt-PD Shaker dimers were meant to clarify the effect of two $VSD^{mut}$ in the channel on the voltage-dependent activation of the $K^+$ conductance.

The nonconductive Shaker dimers ($VSD^{wt}PD^{W434F}$-$VSD^{mut}PD^{W434F}$) exhibit robust gating currents indicating a high level expression of the protein (Supplementary Fig. 2). $Q$–$V$ curves are split into two by the extension of the voltage dependence shift produced by $VSD^{mut}$ (Fig. 5a). Assuming two independent populations of VSDs, wild-type and mutant, we successfully fit the curves with two equally weighted two-state

models, generating $V_0^*$ and $V_1^*$ for voltage dependencies of the two populations of VSD (see "Methods" section and Supplementary Table 2). These fits suggest that VSD movements in the same channel are mostly independent, even when their voltage dependences are not the same. As expected, the variability of $V_0^*$ throughout different nonconductive Shaker dimers is larger than the variability of $V_1^*$ when compared with the two voltage dependences $V_0$ and $V_1$ from nonconductive Shaker channels (homotetramers) (Fig. 5b, c). Since most of the VSD mutations produce negative shifts in the voltage dependence of the sensor, the more negative component of the $Q$–$V$ curves shown in Fig. 5a is the most changed part of the curve when compared with the $Q$–$V$ curve from channels with wild-type VSD only (WT label in the figure). On the other hand, as the second component of the $Q$–$V$ curves is supposed to derive from charge movements from wild-type VSDs in the dimers, their variability is much smaller.

**Intersubunit cooperativity and Shaker dimers**. The wt-PD Shaker dimers $VSD^{wt}PD^{wt}$-$VSD^{mut}PD^{wt}$ are also sensitive to the $VSD^{mut}$ exhibiting shifts in their voltage dependence of the $K^+$ conductance activation. The $G$–$V$ curves were well fitted using a two-state model, yielding a voltage-dependent parameter $V_{1/2}^*$ with small residuals (Fig. 5d and Supplementary Table 2). Interestingly, there is heterogeneity on the $V_{1/2}^*$ shift induced by the presence of a certain $VSD^{mut}$ on wt-PD Shaker dimers (two $VSD^{mut}$ per channel) relative to the shift in $V_{1/2}$ induced by the same $VSD^{mut}$ in Shaker (four $VSD^{mut}$ per channel). This relative parameter (% shift in Shaker) goes roughly from 20 to 80% and on average it is 49.8% (Fig. 5e), suggesting that for the PD open probability, both the voltage dependences from $VSD^{mut}$ and $VSD^{wt}$ present in these channels are important, but it differs in

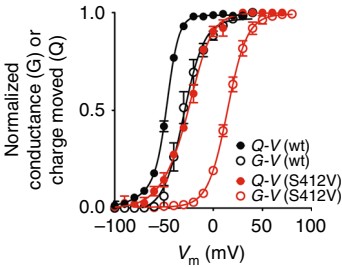

**Fig. 6** Putative key residue S412 in the VSD//PD interface in Shaker. The graph shows $Q-V$ curves recorded in nonconductive Shaker, as well as $G-V$ curves from Shaker carrying (red circles) or not (black circles) the mutation S412V, supposedly disruptive of the VSD//PD interface. $Q-V$ curves are plotted as filled circles and $G-V$ curves with open circles. $Q-V$ curves were fitted with a three-state model to obtain $V_0$ and $V_1$ voltage dependences. $G-V$ curves were fitted with a two-state model to get a midpoint $V_{1/2}$. All data points are the average of 4–6 independent experiments. The vertical bars are the standard error of the mean. Continuous lines over the data points represent the best fit of a three-state model for $Q-V$ curves and a two-state model for $G-V$ curves. All $V_0$, $V_1$, and $V_{1/2}$ voltage-dependent parameters are shown in Supplementary Table 2

separate channels. This matter was not intended to be investigated in this study and therefore will be subject of future studies.

**A putative link in the VSD//PD interface**. Taken together, our data point to a functionally active VSD//PD interface between domains from different subunits of Shaker channel. This interface represents a putative allosteric connection where the voltage dependence of the VSD is transduced into PD movements to generate $K^+$ conductance. This coupling takes effect in addition to the well-described connection between VSD and PD, the S4–S5 linker in the same subunit. Based on the structural models for Shaker channels, PDB:3LUT, there are many amino acids interacting one-on-one at the VSD//PD interface that could be part of this putative functional interface. This interface is expected to be dynamic, since during voltage sensing the VSD residues are hypothesized to move along a fixed rail of side chains from residues along S5, at the PD. We found residue S412, a much conserved residue in S5 of many voltage-dependent $K^+$ channels families, as a possible candidate for this interface. When non-conservatively mutated into valine, the $Q-V$ curve from the double mutant Shaker S412V W434F, together with the $G-V$ curve from Shaker S412V (Fig. 6 and Supplementary Table 3) suggest a VSD-to-PD decoupling, consistent with interference on a putatively functional VSD//PD interface. Indeed, this residue may be only one of several other possible candidates that affect the coupling of the VSD//PD interface.

## Discussion

A salient feature of voltage-gated ion channels is that their open probability is controlled by the membrane potential. What makes this process mechanistically interesting is its high sensitivity to voltage, as much as an e-fold increase in conductance per 2 mV. The exact mechanisms that couple the sensor to the gate(s) are still under debate. Functional and structural data point that the S4–S5 linker is definitely part of the coupling mechanism. In addition, there are strong evidences suggesting the VSD-to-PD coupling mechanisms include noncanonical coupling pathways[8,9,18]. A strong candidate for the noncanonical coupling is the interface between the VSD and the PD (VSD//PD interface). Because these two domains are swapped in the structure, we developed a tandem dimer strategy to evaluate the role of VSD//PDs interface in channel gating. By choosing which VSD and PD

was modified in the dimer, we created channels with pairs of $VSD^{mut}$ and $PD^{W434F}$ in two out of four existing interfaces per channel. Near configuration channels feature two interfaces formed by mutant domains and two formed by wild-type domains. Far configuration channels bear four heterogeneous interfaces. Our data suggest that in Shaker channel the VSDs provide voltage dependence to the PD by a coupling mechanism at the VSD//PD interface. This noncanonical coupling mechanism complements the role the canonical coupling mechanism via the S4–S5 linker.

The dimers make functional channels similar to homo-tetramers. Shaker dimers with wild-type subunits are functionally very similar to homotetramer Shaker (Fig. 1). Channels assembled with W434F-Shaker dimers with only wild-type VSDs show different voltage dependencies in the inactivation process and also different inactivation kinetics according to the relative position of the W434F mutation in the dimer (see Fig. 1). These differences might raise from W434F mutations placed in different domains and not necessarily due to the dimerization per se. Results show that the W434F mutation acts differently in enhancing C-type inactivation when in protomer 1 as compared with protomer 2 and we hypothesize the difference might be due to a restricted protomer 1, since its C-terminal of the PD is bound to the N-terminal of the protomer 2. Conversely, protomer 2 has a free C-terminal. We believe C-type inactivation may be changed by the mobility of the PD C-terminal (Supplementary Fig. 1). Differences appear in the inactivation process and not in activation. The fact that the PD of protomer 1 is part of a VSD//PD interface formed within the same protein, whereas the PD of protomer 2 makes VSD//PD interface between domains from a different proteins, might also explain these differences. Indeed, various channels naturally expressed as multidomain proteins such as K2P[19,20], TPC[21], $Na_V$[12], and $Ca_V$[11] all show asymmetries in structural and functional studies, which are normally attributed to the heterogeneity of the domains (subunits).

Our data could provide insights on function studies if asymmetries around the conductive pore of ion channels are generated by the assembly of the dimers. The difference between far ($PD^{W434F}$ in protomer 2) and near ($PD^{W434F}$ in protomer 1) configured W434F-Shaker dimers containing $VSD^{mut}$, however, is in the inactivation process which we suggest is because part of the inactivation is independent of the open state (different from classical C-type inactivation)[17]. Here, we found that in the channels configured as far, the inactivation curve is monotonic no matter where is the W434F mutation, either in protomer 1 or in protomer 2 of the dimers (Fig. 3c).

$VSD^{ILT}$ is a voltage sensor that moves in two steps: a first step associated to preopening transitions and a second step correlated with channel opening. This feature was used to probe pore inactivation at voltages where a large fraction of the charge of $VSD^{ILT}$ has moved but the pore remains closed. Our results show that pore inactivation develops at those voltages that do not open the pore, i.e., there is inactivation without prior macroscopic ionic conductance. These results revealed what could be two types of pathways to inactivation: the well-known path that requires channel opening and a second independent of channel opening. Inactivation of the near configuration W434F-Shaker dimers (containing $VSD^{ILT}//PD^{W434F}$ interfaces) shows a clear component that develops tens of millivolts more negative than the voltage that opens the channel. The normal inactivation component related to the open channel is shifted to the right, consistent with the activation of the conductance. However, the inactivation component independent of channel activation remained almost unchanged, similar to the inactivation in dimers with only wild-type VSDs (Fig. 2). The remarkable difference between results from far versus near configuration W434-Shaker

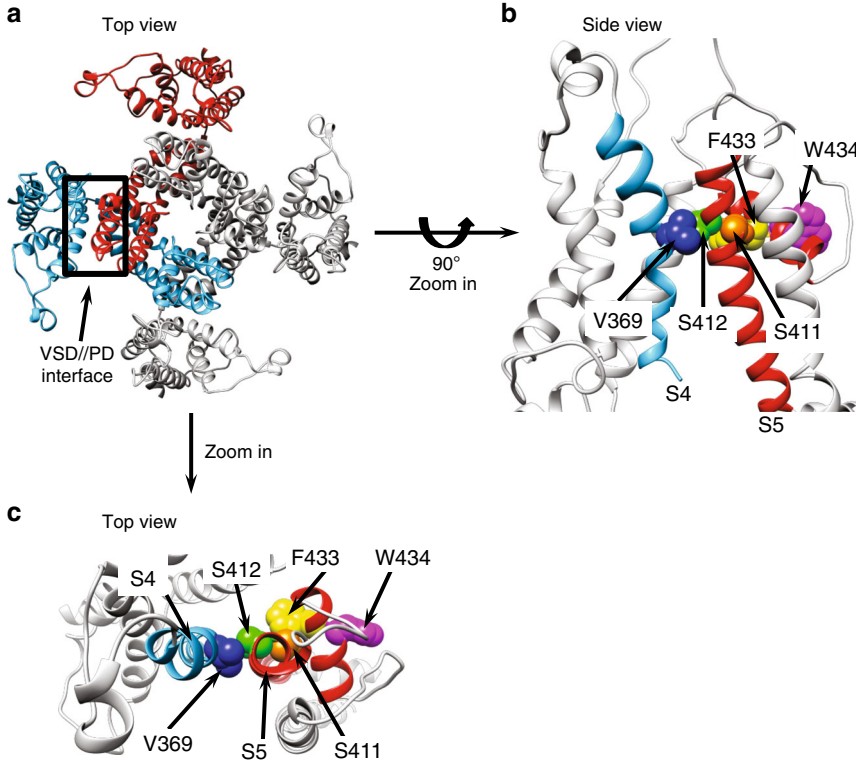

**Fig. 7** Structural models of a Shaker-like channel ($K_V1.2$, PDB:3LUT), showing the suggested VSD//PD interface. **a** Top view of the tetrameric channel, with two subunits in blue and red colors and the other two, for clarity, in light gray. Notice that the black square region indicates the VSD//PD interface between the VSD from different blue and red neighboring subunits. **b** Black square region from **a** zoomed in and turned by 90°. The cartoon shows with ribbons the S4 segment from one subunit in blue and the S5 segment from the neighbor subunit in red. Colored spheres are the van der Waals volumes of the atoms from the residues indicated (V369 in blue, S412 in green, S411 in orange, F433 in yellow, and finally W434 in magenta (residues numbering as in Shaker)). These residues are proposed to be the physical connection of the noncanonical coupling pathway between VSD and PD. **c** Top view of the zoomed black square region shown in **a**. Color code and labels correspond to **a**. Note that in **a** and in **b** the van der Waals surfaces are in contact, giving support to the hypothesis of the VSD//PD interface

dimers rules out the possibility of alternative channel assembling, such as a speculative diagonal arrangement, with the two dimers proteins in a channel crossing each other. In this scenario the resultant channels formed by both near and far configuration W434F-Shaker dimer would form channels with essentially no difference between the relative position of mutants VSD and PD no matter where mutants and wild-types VSDs and PDs are located in the dimer. In this exercise, all channels have the same relative disposition between domains, with four different VSD//PD interfaces (Supplementary Fig. 3).

Data indicate that the inactivation induced by the W434F mutation nearby the selectivity filter is not quite the same as C-type inactivation since the first seems to have two voltage-dependent determinants, which are the voltage sensor movement itself and channel opening. The bacterial $K^+$ channel KcsA, which has no VSD, shows inactivation after activation, a mechanism proposed at the selectivity filter involving water movement[22]. This mechanism may be similar in Shaker, for the pore-opening-dependent inactivation. However, the VSD in Shaker may anticipate the inactivation of the pore during depolarizations, by a noncanonical mechanism.

In wild-type Shaker channels, as opposed to ILT-channels, the two inactivation processes may take place almost simultaneously as voltage is depolarized, appearing indistinguishable. By analyzing the $Q–V$ curves as two sequential processes with two voltage dependences $V_0$ and $V_1$ we were able to verify that the inactivation curves are considerably more affected by shifts in $V_0$ than in $V_1$ but only in the near configuration channels. Consistent

with the above results, we found that the inactivation curve was split into two processes and the more negative component is independent of channel opening.

Our data also revealed an unexpected change in the activation ($G–V$) curves of the W434F-Shaker dimers. The voltage dependence of the $VSD^{mut}$ (measured from nonconductive Shaker) is more effective in modifying the voltage dependence of the voltage-activated $K^+$ conductance in near than in far configuration channels (Fig. 4d, e). This raises the intriguing question: is a W434F-containing PD part of the VSD-to-PD coupling mechanism? We do not have data to confirm this idea but near configuration channels respond much more to a $VSD^{mut}$ than the far configuration ones, suggesting the activation pathway may include the VSD//PD interface, as well as the S4–S5 linker. Future experiments with fluorescent probes reporting at the activation gate may help clarify this possibility. Eliminating the canonical connection by splitting apart the S4–S5 linker in a previous assembled channel may be an obvious direction to move to, although these experiments seem quite challenging.

In nonconductive Shaker dimers it was possible to record robust gating currents from all constructs, and the $Q–V$ curves were split into two. $Q–V$ curves from these constructs originate from pooled data of equal amounts of mutant and wild-type VSDs. The component related to the $VSD^{mut}$ appear in the more negative component (represented by $V_0^*$) of the $Q–V$ curve, since most of the $VSD^{mut}$ in homotetramer Shaker have negative shift in their voltage dependence ($V_0$). The second component of the curve ($V_1^*$) is mostly

made by the currents from the wild-type VSDs. Yet, regardless of whether they are mostly wild-type gating currents, $V_1^*$ was also slightly affected by the VSD$^{mut}$.

The conservative conclusion is that different VSDs in the same channel move independently. Still, considering possible cooperativity, we looked at the voltage dependence of the K$^+$ conductance in wt-PD dimers bearing VSD$^{mut}$ and VSD$^{wt}$ (two each per channel). Altogether, these data suggest that the last step of $Q–V$ curve to open the channel is highly cooperative.

We also identified S412 as a key residue in the Shaker VSD// PD that upon nonconservative mutation greatly modifies the VSD-to-PD coupling (Fig. 6). We point it as a putative key residue not only important to the coupling mechanism in general, but also a key residue to the coupling mechanism that involves the noncanonical pathway, the VSD//PD interface.

Voltage-gated K$^+$ channels can be classified into two groups: domain-swapped and non-domain-swapped channels. In domain-swapped Shaker there are four simultaneous canonical and noncanonical connections between VSD and PD. Our data indicate that the VSD-to-PD coupling in Shaker includes both canonical and noncanonical mechanisms and since they have similar voltage dependences in the wild-type channel, they cannot be separated unless different VSDs are controlling them. Our data also support the idea that W434F is part of the noncanonical VSD-to-PD connection in a network that noncovalently connects the VSD to the PD. We hypothesize that the connection from S4 to the P-loop, where W434F mutation is, includes: V363, I366, or V369 in S4; I409, S411 and S412 in S5; W433 and W434 in the P-loop. Residue S412 appears to be one of the most important elements in the VSD//PD interface because its mutation drastically affects the voltage dependence of the activation process. Upon voltage changes, S4 moves and S412 is thought to be a fixed point for a sliding S4 that uses a residue every three positions to connect S412: V363, I366, or V369 from resting to active states of the VSD, respectively. Therefore, we propose that the main connection between VSD and PD is mediated by a close contact between a residue in the VSD (V363, I366, or V369) and a putative key residue in S5, S412. The residue S412, in turn, contacts the chain made by S411, F433, and then W434F as shown in Fig. 7. All these residues are highly conserved in domain-swapped channels. Previous studies[23] indicate that the interaction S1–PD is functional and we believe this is also part of the VSD//PD interface we are studying here.

Residue S412 is thought to modify the pore gating when mutated by alanine or tryptophan, but the lack of structural studies and VSD charge movement recordings[24,25] prevented accurate rationalization of the effect. Not surprisingly, natural variants of KCNA1, KCNQ1, and KCNQ2, (K$_V$1.1, K$_V$7.1, and K$_V$7.2, respectively), predicting different amino acid in the S412 position (Shaker numbering), are associated with well-known diseases involving cells excitability such as episodic ataxia, long QT syndrome, and epileptic encephalopathy[26–28]. These findings suggest a key residue for channel gating has been mutated. Therefore, we tentatively propose S412 and the abovementioned chain in the S5-pore region of Shaker as a prototype for the noncanonical connection between VSD and PD, with consequences for the activation and inactivation, as a subject of future work.

In conclusion, our results point to the possibility that the noncanonical VSD-to-PD coupling described here is perhaps as important as the canonical mechanism to open and inactivate Shaker. A functional VSD//PD interface transduces voltage dependence to the PD in addition to the classical S4–S5 linker connection between VSD and PD. The voltage dependence elicited by the VSD//PD interface appears in the inactivation, as well as in the activation processes of the PD. Our data contribute to

advance the knowledge on the different ways that the voltage sensor is coupled to the conductance, a basic question in voltage-gated ion channels.

## Methods

**Mutagenesis.** Shaker zH4 K$^+$ channel was cloned into pBSTA expression vector (see full sequence in Supplementary Note 2), a plasmid, optimized for expression in Xenopus oocytes. The optimization included β-globin sequences flanking the Shaker-encoding region, proper Kozac sequence and a poly-A at the 3′ end, just before the linearizing unique restriction site (see below). Shaker cDNA was modified to eliminate fast inactivation of the expressed channel, by a deletion of codons for 41 residues in the N-terminal, delta 6–46[29]. All additional mutations were implemented as needed by PCR (QuickChange; Agilent Technologies) using mismatch mutagenesis primers (Integrated DNA Technologies) (see all primers used in Supplementary Note 3). PCR products were used to transform XL-gold cells (Agilent Technologies) and therefore to produce a sufficient amount of plasmid containing desired cDNA after purification. All mutant plasmids were sequenced to confirm the presence of intended mutations and linearized by digestion with endonuclease NotI (#R3189L New England BioLabs, Ipswich, MA).

**Dimerization strategy.** For Shaker dimers we modified pBSTA containing Shaker-encoding cDNA in two different ways. First we replaced the stop codon TGA in the Shaker sequence by the CCTAGG bases in order to both eliminate the stop codon and introduce a unique restriction site for digestion with AvrII endonuclease (#R0174L New England BioLabs, Ipswich, MA; see primers in Supplementary Note 4). Other additional mutations in the Shaker sequence were also introduced at this point in time. This modified plasmid will contribute with protomer 1 of the Shaker dimers. Second, another plasmid was modified by replacing the start codon ATG by TAGG bases in order to also introduce a restriction site for digestion with AvrII endonuclease and to keep the Shaker open reading frame in the correct frame (see primers in Supplementary Note 4). Similarly, other additional mutations were introduced at this point in the cDNA production line. This second type of modified plasmid will contribute with protomer 2 of the dimer.

Both modified plasmids were digested with AvrII and NotI, the latest a restriction site at about 230 bases away from the stop codon (5′−3′) in our pBSTA. Digested cDNAs, protomer 1- and protomer 2-related, were purified from native agarose gel 5.1 and 1.8 kbp bands, respectively. The protomer 2-related cDNA was then incubated with Antarctic Phosphatase (AP, #M0289S New England BioLabs) for 5′-ends dephosphorylation. Finally, protomers 1- and protomer 2-related (AP-dephosphorylated) purified cDNA were mixed together in 1:3 molar ratio for DNA ligation with T7 DNA Ligase (New England BioLabs #M0318). The ligation mix was used to transform XL-gold cells (Agilent Technologies) and therefore produce a sufficient amount of plasmid containing desired cDNA after purification. In the case of Shaker dimers production line, all purified cDNA from bacteria were digested with NotI for linearization and a small sample of it was run in 2% native agarose gel to confirm dimerization by the detection of a band at 7 kbp size. After confirming dimerization the samples were submitted to sequencing to confirm mutations were introduced in right positions and also that unwanted mutations were absent. In the sites we introduced mutations in the VSD and/or PD regions we observed a double peak in the chromatogram, strongly suggesting heterodimers.

All linearized plasmids were then transcribed in vitro (T7 RNA expression kit, Ambion Invitrogen, Carlsbad, CA) from a T7 promoter region where the transcription of the Shaker cDNA starts in the pBSTA plasmid version used here. The cRNA synthesized by this procedure were then precipitated and resuspended in RNAse-free water for injections in to Xenopus oocytes.

**Channels expression in Xenopus oocytes.** Female Xenopus laevis were purchased from Nasco (Fort Atkinson, WI) and housed in the Animal Resources Center of the University of Chicago, in accordance with animal usage protocol #71475 of the University of Chicago Institutional Animal Care and Use Committee—IACUC. Usually 12–24 h after oocytes surgical extraction from adult frogs, 10–50 ng of cRNA were injected into defoliculated oocytes in 50 nl RNAse-free water. Before and after being injected, oocytes were sitting in glass petri dishes containing standard oocyte solution with the following composition: 100 mM NaCl, 5 mM KCl, 2 mM CaCl$_2$, and 10 mM Hepes, pH 7.5, supplemented with 50 μg/ml gentamycin. Injected oocytes were kept for 1–3 days from injection, at 18 °C.

**Electrophysiology.** Gating and ionic K$^+$ currents were recorded from oocytes using the cut-open voltage-clamp method[30] at room temperature (22 °C). The resistance of a voltage-measuring micropipette placed inside the oocyte for the virtual ground feedback was 0.3–0.8 MΩ. Transient capacitive currents were minimized from the recorded currents by a dedicated circuit and the remaining currents plus the linear leak were subtracted online using the P/N method when possible[31]. Current data were filtered online at 10–20 kHz using a low pass Bessel filter in the voltage clamp amplifier (CA-1B, Dagan Corporation, Minneapolis, MN, USA), digitized at 16-bits and sampled at 50–100 kHz (Innovative integration). Data were stored and analyzed using in-house software. All gating current recordings were made with the background mutation W434F, except when

indicated. The internal gating current recording solution was 120 mM N-methyl-D-glucamine (NMG) methylsulfonate (MES), 2 mM EGTA, and 10 mM Hepes, pH 7.5, and the external solution contained 120 mM NMG-MES, 2 mM Ca-MES, and 10 mM Hepes, pH 7.5. All $Q$–$V$s analysis included 4–6 independent experiments. Charge–voltage ($Q$–$V$) curves were fitted with a three-state model as described previously, yielding two VSD voltage-dependent parameters, $V_0$ and $V_1$, and their apparent charges $Z_0$ and $Z_1$, respectively[15,32]. For ionic current recordings, the internal solution contained 120 mM K-MES, 2 mM EGTA, and 10 mM Hepes, pH 7.5, and the external solution contained 12 mM K-MES, 108 mM NMG-MES, 2 mM Ca-MES, and 10 mM Hepes, pH 7.5. All ionic currents analysis also included 4–6 independent experiments. Conductance–voltage ($G$–$V$) curves were obtained by plotting the maximal K$^+$ conductance at each depolarizing voltage step. $G$–$V$ curves were fitted to a single two-state function, yielding a PD voltage-dependent parameter $V_{1/2}$, a value that denotes the voltage for half-maximal macroscopic K$^+$ conductance activation. For all experiments with conductive Shaker channels a standard holding potential of −80 mV was used and a conditioning period of time at hyperpolarized voltage (value varied according to the mutant) was used to insure channels were closed before a depolarizing voltage was applied.

K$^+$ currents recorded in all types of Shaker dimers were activated by a depolarizing voltage and the currents inactivated with a time constant of seconds. Every 30 s cycle period, from a holding potential of −90 mV and after a 100 ms conditioning pulse at −120 mV, a 10 s depolarizing pulse to a different voltage (from −110 to +60 mV, every 10 mV, or otherwise stated) was followed by a 5 ms pulse to −120 mV and a 1-2 sec pulse to +60 mV (or otherwise stated). The first pulse at different voltages along the series activates K$^+$ currents in a voltage-dependent manner and inactivates it in a time-dependent manner afterwards. The second pulse, fixed to +60 mV, tests how much of the current was inactivated by the 10 s first pulse at a given voltage (see pulse diagram in Fig. 2a).

**Data analysis and statistics**. In-house software was used to analyze the data and GraphPad Prism (GraphPad Software, Inc) was used to fit the data with equations (see below).

In nonconductive channels or otherwise stated, gating (sensing) currents activated by a depolarizing voltage step were integrated in time for its total charge. $Q$–$V$ curves from wild-type Shaker cannot be fitted with a two-state model. Instead, a three-state model is necessary to fit the data, with at least one intermediate state between resting and active states. It is important to note that the equation used to fit the data is derived from a model that includes three VSD states in sequence (Eq. (1)):

$$Q(V_\mathrm{m}) = N \frac{Z_1 + Z_0 \left(1 + \exp\left(-Z_1 e_0 \frac{(V_\mathrm{m} - V_1)}{kT}\right)\right)}{1 + \left(\exp\left(-Z_1 e_0 \frac{(V_\mathrm{m} - V_1)}{kT}\right)\right)\left(1 + \exp\left(-Z_0 e_0 \frac{(V_\mathrm{m} - V_0)}{kT}\right)\right)}, \quad (1)$$

where $Q(V_\mathrm{m})$ is the charge moved at a given membrane voltage $V_\mathrm{m}$, $N$ is the number of sensors, $Z_0$ and $V_0$ are respectively the charge and the voltage of the first transition, from resting to an intermediate state and $Z_1$ and $V_1$ are, charge and voltage involved in the second transition, from the intermediate to active state, $e_0$ is the elementary charge, $k$ is the Boltzmann constant, and $T$ is the absolute temperature in Kelvin.

Therefore the fitted $Q$–$V$s gave two voltage-dependence-related fitted parameters from VSD: $V_0$ and $V_1$, respectively from the voltage-dependent transition from resting state to the intermediate state and from the latter to active state. Normalized $Q$–$V$ curves from at least four independent experiments were averaged and fitted with the three-state model described above.

In some cases presented here, a split $Q$–$V$ curve is obvious by simple inspection where there is a clear presence of an intermediate state of the mutant VSD in question. However, in most of the VSD mutants, and also in the wild-type VSD the intermediate state is not obvious.

K$^+$ conductance was computed from K$^+$ currents peaks at different membrane voltages following Hodgkin and Huxley[33] equation:

$$G_K = \frac{I_K}{\left(V_\mathrm{m} - \left(\frac{kT}{e_0} \ln \frac{[K^+]_\mathrm{out}}{[K^+]_\mathrm{in}}\right)\right)}, \quad (2)$$

where $G_K$ is the K$^+$ conductance, $I_K$ is the K$^+$ current peak, $V_\mathrm{m}$ is the membrane voltage. $[K^+]_\mathrm{out}$ and $[K^+]_\mathrm{in}$ are the potassium ion concentrations outside and inside.

Conductance–voltage relationships ($G$–$V$ curves) were built by plotting conductance values against the respective membrane voltage they were recorded at. Normalized K$^+$ conductance curves were averaged from at least four independent experiments and fitted with a two-state model (Eq. (3)) for a voltage-dependence parameter $V_{1/2}$ which is the voltage related to a half-maximal activation of the normalized K$^+$ conductance:

$$G_K(V_\mathrm{m}) = \frac{1}{1 + e^{-Z e_0 \frac{V_\mathrm{m} - V_{1/2}}{kT}}}, \quad (3)$$

where $G_K(V_\mathrm{m})$ is the K + conductance at membrane voltage $V_\mathrm{m}$, $Z$ is the apparent charge that cross the electric field and is involved in channel opening, $V_{1/2}$ the voltage dependence for the open channel.

Inactivation curves (Inact-$V$) were built by plotting the peak of the K$^+$ current at +60 mV membrane voltage, after a period of 10 s at different membrane voltages. After normalization and averaged from at least four independent experiments, Inact-$V$ curves were fitted with a two-state model (Eq. (4)):

$$\mathrm{Inact}(V_\mathrm{cp}) = 1 - \left(\frac{1}{1 + e^{-Z e_0 \frac{V_\mathrm{cp} - V_\mathrm{inact}}{kT}}}\right), \quad (4)$$

where $V_\mathrm{cp}$ is the membrane voltage during the conditioning period, $Z$ is the apparent charge that cross the electric field involved in inactivation process, $V_\mathrm{inact}$ is the voltage dependence of the inactivation process.

Some Inact-$V$ curves were not well fitted by Eq. (4), generating big residues as measured by the absolute sum of the squared residues. In these cases the Inact-$V$ curve were split in two components and therefore we used a model including two independent transitions, assuming two independent voltage-dependent inactivation processes. The equation in this case was Eq. (5):

$$\mathrm{Inact}(V_\mathrm{cp}) = 1 - \left(\frac{A_1}{1 + e^{-Z_1 e_0 \frac{V_\mathrm{cp} - V_{1st}}{kT}}} + \frac{1 - A_1}{1 + e^{-Z_2 e_0 \frac{V_\mathrm{cp} - V_{2nd}}{kT}}}\right), \quad (5)$$

where $A_1$ is the fractional amplitude of the first inactivating component, $V_\mathrm{cp}$ is the membrane voltage during the conditioning period, $Z_1$ and $Z_2$ are the apparent charges involved in the first and second component of the inactivation curve, respectively, $V_{1st}$ and $V_{2nd}$ are the voltage dependence of the first and second components, respectively.

**Reporting summary**. Further information on research design is available in the Nature Research Reporting Summary linked to this article.

## Data availability

The data that support the findings of this study are available on reasonable request from the corresponding author. The values behind Figs. 1, 3–6 are provided as Source Data File.

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

## Acknowledgements

We thank Drs. E. Perozo and M. Holmgren for their careful reading and their suggestions and comments on the manuscript. Supported by NIH grant R01-GM030376.

## Author contributions

J.L.C.-d.-S. and F.B. conceived the project. J.L.C.-d.-S. performed research, analyzed data and wrote the manuscript, receiving inputs from F.B.

## Additional information

**Competing interests:** The authors declare no competing interests.

