## [Peer Review File · Nature Communications]

Reviewers' comments:

Reviewer #1 (Remarks to the Author):

This manuscript describes experiments that aim to interrogate the role of the non-canonical interface between voltage-sensing domains and pore domains in the Shaker Kv channel. In most Kv channels, the S4 helix within the voltage-sensing domain of one subunit is positioned next to the S5 helix within the pore domain of an adjacent subunit. The canonical region implicated in coupling voltage sensor activation with opening of the pore is the S4-S5 linker helix positioned parallel to the membrane and enabling the domain-swapped architecture. More recently, studies have begun to study the non-canonical interface between the S4 of one subunit and the S5 helix of the adjacent subunit. In the present study the authors construct tandem dimers and then insert the W434F mutation into the pore domain of either the first or second protomer, and then introduce any one of a number of mutations into the voltage-sensing domain of either the first or second protomer based on previous studies. The basic finding is that the conductance (G)-voltage (V) and steady-state inactivation relations are different when mutants are present in the first or second protomer, which the authors interpret to mean that the non-canonical interface between the S4 of one subunit and the S5 of the adjacent subunit is critical for coupling voltage-sensor activation with pore opening or inactivation. Although I think the authors may be onto something, and the data presented are likely to have been collected carefully, the presentation does not do a good job of explaining what boils down to a quite complex story. Part of the problem is the authors rush through complex results with hard to understand nomenclature, and then try and flesh everything out in a super long discussion where they present additional results. To my thinking, this paper needs to be carefully re-written from scratch after coming up with easier to understand nomenclature and walking the reader step by step through each of the results and drawing clear and cautious interpretations at each step. The discussion should then fit those conclusions together with the larger body of work and point the way forward to understanding the coupling mechanism in more depth. It is hard to assess whether this work is interesting and important until the authors have done a much better job of presenting their work and ideas. The authors also must address what may be a fatal flaw in the work, which is that the assembly of dimers into a dimer of dimers is not faithful and is mutation-dependent. I can't help but worry that some of the differences seen here are a consequence of unfaithful assembly.

A few more specific comments:

1) The authors should improve the cartoons in Fig 1, come up with a better nomenclature to describe the various constructs and restate in other words as often as they can tolerate to make sure the reader can follow. In Fig 1 it would help to show the physical connection between the C-terminus of the pore domain of one subunit and the voltage sensor of the adjacent subunit in the dimer. I also think it might help to always have the first protomer white and the second gray, and then to indicate the presence of mutants in either the voltage sensor or the pore with a red asterisk. It doesn't make sense to give the near and far constructs distinct names until they harbor mutations, because at least if I understand the basic paradigm, there is only once tandem dimer construct.

2) The control data with only W434F introduced into the first or second protomer that is described lines 112-127 (please use page numbers!) and shown in Fig 2 seems like a potentially big problem. If everything was assembling properly, I can't think of any mechanism to explain the observed difference. The mostly likely explanation is that the W434F mutant is not as frequently incorporated into the dimer of dimers when present in the second protomer. If this is happening, wouldn't this greatly complicate the interpretation of the remaining data?

Reviewer #2 (Remarks to the Author):

In voltage dependent activation of ion channels the interactions between the VSD and PD that couple the voltage sensing to the pore opening are important but not yet fully understood. In this manuscript, using a creative approach of tandem dimers SHAKER constructs (td-SHKnear and td-SHKfar) that can separate the different VSD-PD interfaces, the authors aim to reveal that an inter-subunit interaction between residues in the VSD and residues in the PD is important for the VSD-PD coupling. The comparison of td-SHKnear and td-SHKfar show some interesting results. For example, in the td-SHKnear case, the pore opening (V1/2 from GV curve) correlates with the second component of the VSD movement (V1 from QV); and the first inactivation component (AE1) correlates with the first component of the VSD movement (V0 from QV). whereas the second inactivation component (AE2) follows the V1 as well as the V1/2. In addition, the slope of td-SHK V1/2 vs V1/2 of SHK, V0 and V1 provide the major argument for the inter-subunit "near" interaction to be important for VSD-PD coupling. Some concerns are as follows.

1. The tandem dimers are an excellent approach to study the VSD-to-PD coupling in SHK channels in order to dissect the "near" and "far" interactions. Since this approach is likely to become a standard approach in future studies of the VSD-PD coupling, some aspects of this approach need to be validated and developed further. These include:

A) The authors state "a close proximity between VSD and PD from neighbor protomers is obvious". However, the distance between neighbor and diagonal protomers does not seem to be that different. The possibility of diagonally arranged tandems, and other possibilities of arrangements among VSDs and PDs than the ones depicted in Fig 1 should be excluded experimentally.

B) Fig 2. Why do td-SHKN-ter and td-SHKC-ter have different inactivation properties? Apparently, the tandem constructs created asymmetry in the tetrameric channel. This asymmetry weakens any conclusion drawn from the experiments using these tandem dimers.

C) The allosteric effects on the "far connection" need to be excluded when the "near connection" is considered.

2. The manuscript compares properties of td-SHKnear and td-SHKfar against a number of other forms of the channel. The conclusions are derived from either the slope of the comparison (e.g. Fig 4) or r2 e.g. Fig 5, 6). This approach is novel and interesting. On the other hand, the interpretation of the results need to be more clearly developed for readers to follow. These include:

A) The authors may want to explain the rationale of using either slope or r2 , but not both, for interpretation of the results, and describe more clearly the physical significance behind the phenomena. What do the slope and r2 indicate on the VSD-PD interactions?

B) In Fig 4A results, it is not clear why td-SHKfar has a shallower slope than td-SHKwt-intact. Please explain the results based on the comparison of all VSD-PD interactions. This result seems not to support the conclusion that the near interaction is important based on the slope.

C) Line 177. "we find that only V1 had significant correlation". Please define a value of r2 for "significant correlation".

D) In Fig 4, 5, 7, V0 V1 of SHKW434F are compared to properties in td-SHK. The former has four subunits with W434F but the latter has only two. Is the Q-V (V0 and V1) the same in these two sets of channels?

E) The authors seem to conclude that in Fig 8 the two components of Q-V of td-SHKallW434F are derived from two different pairs of VSD. This assignment needs experimental justification. In addition, raw data for Figs 7 and 8 need to be shown.

3. In describing the Fig 3 results the authors state: "The broader range of voltages spanned by G-V and Inact-V curves from td-SHKnear channels resemble the voltage-dependences of the regular SHK bearing those respective mutations in the VSD." What does "resemble" mean here? Since td-SHKnear only contains half of the "near connections" of SHK, is it expected that the results from the two channels should differ?

4. The authors state that the S4-S5 interface is important for the inter-subunit "near" interaction, and show data and discussion of S412 (Fig 9) to support this statement. However, the data in Results focus on the interaction between VSD residues with W434F in the PD. Are these interactions considered important for the inter-subunit "near" interaction, and are they part of the

S4-S5 interface in authors' mind? The addition of the S412 data and discussion is confusing, and it is not a complete study to support the importance of S4-S5 interface in VSD-PD coupling. The evidence for the residue being important in the coupling is not strong and subjects to other interpretations.

5. The manuscript is hard to follow and some revisions may help readers like this reviewer. Some examples are:

A) Fig 3. Add a figure of structure showing all the residues 358, 381 and 434 (probably 412 as well).

B) Fig 4,5. A diagram showing different channels with mutations will be useful. The mutations need to be revealed and labeled in the existing panels.

C) Nomenclature V1/2(1), (2) in the text are different from those in Fig 8.

D) Lines 310 and 311. Use either KCNQ1 (KCNQ2) or Kv7.1 (Kv7.2).

6. The discussion in lines 310-316 on channels other than SHK is speculative and not supported by any analyses.

Minor concerns.

1. Fig 2, panel C mislabeled.

2. Line 155. Check the reference.

3. Line 421, 422: "Fig 6" should be "Fig 7".

Reviewer #3 (Remarks to the Author):

This manuscript addresses the voltage-sensor pore coupling by using tandem dimers and electrophysiology recordings. The main finding of the paper is that the S4/S5 interface controls voltage-dependence of C-type inactivation and transduction of VSD voltage-dependence to the open pore probability.

The question is of broad interest and the approach used could be very interesting to the field but I find it difficult to verify that the claims are substantiated by the data due to quite severe communication problems that obscure the message:

1- The nomenclature of the different constructs and mutants

is unclear. Please provide a table with the different constructs listed, the mutations present and make an attempt to make the names of the constructs informative and consistent. Indeed the nomenclature seems to be shifting along the paper. For example on line 192, V0, V1 and V1/2 is said to be calculated from SHK, Table 1 indicates that V0 and V1 are measured in SHKW434F and V1/2 in SHK while Fig 5.D indicates that V1/2 is evaluated in td-SHK. Sometimes VSDmut is used to indicate a construct, and sometimes to talk about the fact that the VSD is mutated within a construct.

2- The organization of the data is confusing: new experiments are introduced in the discussion section. Section titles do not always reflect the contents of the section, the first result section does not have a title which does not allow me to understand the main point made.

3- Purely descriptive statements about results are not discussed, either in the results section or in the discussion, as far as I can tell. As examples (but many others can be cited): I178: In addition, the V1 in the near case correlated better than the far case (Fig. 4B). End of paragraph. I198: The only strong correlation found was between parameters from td-SHKnear, where the τ_{slow} correlated well with the V1/2 of the conductance activation. End of paragraph. Conversely, seemingly crucial aspects are neither mentioned nor discussed: for example, why are some meaningful correlations positive and others negative?

3- In several instances, the data presented in the text and tables is not compatible with the

figures, as if new data points had been recorded but the numbers not updated consistently. The correlation coefficients in Figure 5 and Table 1 are different, I believe they should represent the same data. In figure 8, V_0 and V_1 are presented, the text talks about $V_{1/2(1)}$ and $V_{1/2(2)}$, which I believe are the same things and the r^2 presented on l.267 is different from the one in Fig.8. These are only examples, the manuscript should be thoroughly checked for consistency. Fig 4.B seem to contain two blue and two black data points at $V \sim -80\text{mV}$, it is difficult to understand why.

4- The variables used for different measurable quantities and model parameters sometimes change between the text, the caption and the graph axes (Fig. 8 for example).

5- The language is also in several instances difficult to understand. As an example, I did not understand the following statements: l171: "These numbers show that in the near dimer case, the presence of two VSDmut influences the voltage-dependence of the channel conductance activation more than what would be expected for the abundance of VSDmut"; l182: "The plot of the first inactivation component's amplitude against V_0 , but not V_1 from VSDmut, gives a linear correlation with $r^2=0.68$ for the case of td-SHKnear and with $r^2=0.20$ for td-SHKfar (Fig. 5A).

I am willing to review a revised version if a serious attempt at clarifying and organizing the information is made because of the relevance and the importance of the topic. It might be a good idea for the authors to give it to a scientist which is not immediately from the field to proof-read it.

Reviewer #4 (Remarks to the Author):

In Shaker channel, and most probably in many other voltage-gated channels, coupling between the voltage sensor domain VSD and the pore domain PD is, at least partly, realized by the linker between the two domains, S4-S5. The role of S4-S5 as a mechanical lever has been suggested in many studies. Another component of the coupling is made possible by the swapped domain arrangement of Shaker, which put the S4 voltage sensor of one subunit in direct apposition to the S5 of the adjacent subunit. In the present manuscript, Carvalho-de-Souza and Bezanilla use tandem dimers of Shaker channel to test if this non-covalent interaction between S4 of one subunit and S5 of adjacent subunit plays a role in the VSD/PD coupling.

The present manuscript provides new perspectives for the non-canonical mechanism of VS/PD coupling, but the results obtained here require additional experiments/analysis.

1. In a previous work published in JGP, the same group has described that a mutation in Shaker PD (W434F) drastically influences the effect of one mutation in the VSD (L361R), but without detailing the molecular mechanism. In particular it is not known if the observed functional interaction is due to the "far connection" between the VSD and PD of the same subunit, and/or the "near connection" of VSD/PD of adjacent subunits. Here, the use of dimer to discriminate the role the two connections seems relevant and elegant, but some observations raise some doubts about the conclusions.

In the td-SHKnear dimeric construct (Figure 1), PD mutation W434F is inserted in the first protomer. In the td-SHKfar construct, the same mutation is inserted in the second protomer. In the first protomer, the "C-terminus" of PD is linked to the VSD of the second protomer, whereas in the second protomer, the "C-terminus" of PD is free. This asymmetry is probably the cause of a major difference in the inactivation properties between the 2 controls (td-SHKN-ter and td-SHKC-ter), that present the W434F mutation in one of the PDs but no mutation in VSD (figure 2). This suggests that intrinsic stability of the PD is different between the 2 controls. Then it becomes impossible to conclude that the differences between the td-SHKnear and SHKfar activation/inactivation curves, observed when a mutation is introduced in VSD (figure 3) are related to the "far connection" or "near connection" between the VSD and PD. It may only be due

the different intrinsic stability of the PD of the 2 constructs.

One way to overcome this first issue, would be to realize the same experiments on a different background: in both constructs the mutated PD would be in the second protomer (PD with the free C-ter). VSD mutation would be in VSD1 for td-SHKnear and in VSD2 for td-SHKfar. In that case, the two controls (no mutation in VSD) may be more similar than the ones presented in Figure 2.

2. Alternatively, the observed influence of W434F on VSDmut may be due to the interaction between adjacent S1 and PD, which has been suggested by statistical coupling analysis (Lee SY, Banerjee A, MacKinnon R. PLoS Biol. 2009; 7(3):e47). The results presented here cannot exclude that the S1/PD interaction is a major component of the VSD/PD coupling. Similarly to the S4/S5 interaction, this interaction may participate to the differences between the td-SHKnear and SHKfar activation/inactivation curves.

3. In the previous work in JGP, the 4 W434F mutations (far+near) in the tetramer induce a right shift in QV curve of L361R. The model would be strengthened if, in K-depleted solution, the 2 "near" W434F mutations also induce a right shift in QV curve of L361R.

Also for 412V mutant, it would interesting to test the variation in coupling using tandem constructs in K-depleted conditions.

4. A major part of the conclusions relies on correlation and r^2 comparisons, but the p-value of the correlation, which is provided by Prism, is never indicated.

Second, in line 181 it is stated that the first component of inactivation is correlated with VSD movement, partly based on r^2 of A1 with V0 (Figure 5A). However, the second component (1-A1) should give the same correlation.

Last, it would be useful if the author justify the use of AE1 and AE2 parameters.

Minor

5. Line 143: a figure would be useful for the comparison.

The manuscript contains many mistakes, some of them, render it difficult to read. Here are some.

1. Line 115 Equation 2, not 1
2. In figure 2, a panel is mislabeled. Figure 2C, not B
3. Line 150, "starts with negative voltages" is not clear
4. Lines 189-192, please explain in more details since data in Figure 5 and table 1 corresponds to different conditions. This is not clear in the present form.
5. Line 198 Figure 6C and D not 5C and D.
6. Line 263. Does V1/2(1) correspond to V0-dimer in figure 8?
7. Line 278 : Figure 4A not 3C
8. Line 345 : were not was
9. Line 349 : delta 6-46
10. Line 370 : "band" is lacking, kbp and not kDa. cf also line 378.
11. Line 372 : dephosphorylation
12. Line 373 : AP-dephosphorylated not digested
13. Line 382 : a promoter does not enhance transcription
14. Line 399: sentence is not clear. "Supplied" should be replaced by "decreased" and "uncompensated current" by "remaining capacitive current"

15. Lines 416-419 are probably inappropriate since another protocol follows in the next paragraph, with a different HP.
16. Lines 436-438: repetition of the same idea.
17. Line 474 averaged
18. Please remove all the 's

Reviewers' comments:

Reviewer #1 (Remarks to the Author):

This manuscript describes experiments that aim to interrogate the role of the non-canonical interface between voltage-sensing domains and pore domains in the Shaker Kv channel. In most Kv channels, the S4 helix within the voltage-sensing domain of one subunit is positioned next to the S5 helix within the pore domain of an adjacent subunit. The canonical region implicated in coupling voltage sensor activation with opening of the pore is the S4-S5 linker helix positioned parallel to the membrane and enabling the domain-swapped architecture. More recently, studies have begun to study the non-canonical interface between the S4 of one subunit and the S5 helix of the adjacent subunit. In the present study the authors construct tandem dimers and then insert the W434F mutation into the pore domain of either the first or second protomer, and then introduce any one of a number of mutations into the voltage-sensing domain of either the first or second protomer based on previous studies. The basic finding is that the conductance (G)-voltage (V) and steady-state inactivation relations are different when mutants are present in the first or second protomer, which the authors interpret to mean that the non-canonical interface between the S4 of one subunit and the S5 of the adjacent subunit is critical for coupling voltage-sensor activation with pore opening or inactivation. Although I think the authors may be onto something, and the data presented are likely to have been collected carefully, the presentation does not do a good job of explaining what boils down to a quite complex story. Part of the problem is the authors rush through complex results with hard to understand nomenclature, and then try and flesh everything out in a super long discussion where they present additional results. To my thinking, this paper needs to be carefully re-written from scratch after coming up with easier to understand nomenclature and walking the reader step by step through each of the results and drawing clear and cautious interpretations at each step. The discussion should then fit those conclusions together with the larger body of work and point the way forward to understanding the coupling mechanism in more depth. It is hard to assess whether this work is interesting and important until the authors have done a much better job of presenting their work and ideas. The authors also must address what may be a fatal flaw in the work, which is that the assembly of dimers into a dimer of dimers is not faithful and is mutation-dependent. I can't help but worry that some of the differences seen here are a consequence of unfaithful assembly.

A few more specific comments:

- 1) The authors should
 - a) improve the cartoons in Fig 1
We thank the reviewer for pointing this out. We made new cartoons for Figure 1.
 - b) come up with a better nomenclature to describe the various constructs and
We appreciate this criticism and we have tried hard to improve the nomenclature of the constructs for a better understanding. The new nomenclature can be found

throughout the whole manuscript. Besides, we included a table (Table 1) that contains a list of all constructs using the new nomenclature, together with pertinent voltage dependent parameters.

c) restate in other words as often as they can tolerate to make sure the reader can follow.

We recognize our wording was too specialized and perhaps not quite adequate for a scientist that is not directly involved in this field. We rewrote the entire manuscript and we gave the new manuscript to colleagues who also made important improvements in the presentation, so that this new version should be easier to follow.

In Fig 1 it would help to show the physical connection between the C-terminus of the pore domain of one subunit and the voltage sensor of the adjacent subunit in the dimer.

The new cartoons show the connection between protomers as suggested – thank you.

I also think it might help to always have the first protomer white and the second gray, and then to indicate the presence of mutants in either the voltage sensor or the pore with a red asterisk. It doesn't make sense to give the near and far constructs distinct names until they harbor mutations, because at least if I understand the basic paradigm, there is only once tandem dimer construct.

We thank the reviewer for suggesting this change. As stated above, now the cartoons are in color for a better identification in Figures 1 and 2. First protomer is shown in blue and the second in red. We also added markers (circles and diamonds) to identify mutations, when present.

2) The control data with only W434F introduced into the first or second protomer that is described lines 112-127 (please use page numbers!) and shown in Fig 2 seems like a potentially big problem. If everything was assembling properly, I can't think of any mechanism to explain the observed difference. The mostly likely explanation is that the W434F mutant is not as frequently incorporated into the dimer of dimers when present in the second protomer. If this is happening, wouldn't this greatly complicate the interpretation of the remaining data?

We acknowledge we were supposed to use page and line numbers. The present version of the manuscript now includes page numbers.

Relative to the likelihood of non-incorporation of the W434F mutation in the dimer, when that mutation is in the second protomer, we believe it is extremely low. We respectfully hold this position because, relative to the regular Shaker, i) the inactivation rates in W434F-containing Shaker dimers are crucially faster and ii) the relative amplitude of the fast component of the inactivation process is increased no matter where is the W434F mutation, if in protomer 1 or 2. As factual arguments we should add that, first, all cDNA constructs were fully sequenced before *in vitro* transcription for confirmation of mutations (please see methods section for details). Second, from a functional point of view, either when W434F mutation is placed in protomer 1 or 2, the

inactivation parameters are very different from Shaker dimers that do not contain W434F mutation: the faster component is even faster (Fig. 1f) and its relative amplitude is higher (Fig. 1g). This indicates W434F mutation was always incorporated. The faster inactivation process in W434F-containing Shaker dimers is also reflected in the available currents after 10 seconds depolarization voltage periods (Fig. 1h).

Although our conclusions are mainly based on the voltage dependence than on time constants, we have considered reasons for the differences observed in the time constants. When W434F mutation is in the second protomer it seems that it is not as effective as it is when in the first protomer in speeding up inactivation. We discuss this issue in the new version of the manuscript in the paragraph starting with the subtitle "*The dimers make functional channels similar to homotetramers*", line 433 in the discussion section, where we give possible reasons for the differences. We believe, that the main reason of the source of asymmetries is illustrated in Fig. S1: the C-terminal status, free or bound, of the PD^{W434F}.

Despite the above differences between W434-Shaker dimers with the W434F mutation in protomer 1 and 2, we believe our data is clear in that it shows how the inactivation process is controlled by the VSD that is "near" the pore domain containing the W434F mutation. Therefore, to further clarify the issue, a new set of experiments were done and it is shown in the new Fig. 3 as new data added to the manuscript. This new data show that regardless where W434F mutation is (also see Fig. 2 for related cartoons), either in protomer 1 or in protomer 2, the inactivation curve is consistently split when the mutant VSD is "near" the pore domain containing W434F mutation. Once again, it is also important to note that in the Methods section we describe our technique that insures we have dimers and that they have the mutations in the proper positions.

Reviewer #2 (Remarks to the Author):

In voltage dependent activation of ion channels the interactions between the VSD and PD that couple the voltage sensing to the pore opening are important but not yet fully understood. In this manuscript, using a creative approach of tandem dimers SHAKER constructs (td-SHKnear and td-SHKfar) that can separate the different VSD-PD interfaces, the authors aim to reveal that an inter-subunit interaction between residues in the VSD and residues in the PD is important for the VSD-PD coupling. The comparison of td-SHKnear and td-SHKfar show some interesting results. For example, in the td-SHKnear case, the pore opening ($V_{1/2}$ from GV curve) correlates with the second component of the VSD movement (V_1 from QV); and the first inactivation component (AE1) correlates with the first component of the VSD movement (V_0 from QV). whereas the second inactivation component (AE2) follows the V_1 as well as the $V_{1/2}$. In addition, the slope of td-SHK $V_{1/2}$ vs $V_{1/2}$ of SHK, V_0 and V_1 provide the major argument for the inter-subunit "near" interaction to be important for VSD-PD coupling. Some concerns are as follows.

1. The tandem dimers are an excellent approach to study the VSD-to-PD coupling in SHK channels in order to dissect the "near" and "far" interactions. Since this approach is

likely to become a standard approach in future studies of the VSD-PD coupling, some aspects of this approach need to be validated and developed further. These include:

A) The authors state “a close proximity between VSD and PD from neighbor protomers is obvious”. However, the distance between neighbor and diagonal protomers does not seem to be that different. The possibility of diagonally arranged tandems, and other possibilities of arrangements among VSDs and PDs than the ones depicted in Fig 1 should be excluded experimentally.

We agree this is an important question that needs to be clarified, thanks! Indeed, when we say a VSD is “near” a PD to form a VSD//PD interface we mean they are a few angstroms away from each other, if we measure the distances between alpha carbons of residues from S4 and from S5 (from different subunits). For example, taking one VSD as reference, there is only one PD that is very close to it which is the one forming the interface (please see Fig 7). The other PDs in the channel are far from that VSD, therefore they are not making direct contact with that particular VSD. We do believe a diagonal arrangement of the protomers of the dimerized Shaker, although theoretically possible, is not the case for the present data. Nevertheless, if we speculate diagonally arranged protomers with cartoons (please see the figure below), the results are channels with essentially no difference between the relative position of mutants VSD and PD no matter where mutant and wild type VSD and PD are located in the dimer. In this exercise, all channels have the same relative disposition between domains, with four

different VSD//PD interfaces.

Fig R1. Speculative diagonally arranged dimerized Shaker proteins. **a** and **b** show how the channels would look from the extracellular side if the mutations in VSD and PD are in different protomers as for *near* channels configuration. **c** and **d** show top views of channels putatively formed by proteins that were mutated in their VSD and PD in the same protomer as for *far* channels configuration. Note that the four channels shown in **a-d** are very similar in their VSD//PD interfaces, with the same four different interfaces in all of them.

The data in Fig. 3 show plots remarkably different between *near* and *far* configured Shaker dimers, showing that the channels must be arranged most preferably with protomers side-by-side as opposed to diagonally arranged. For example, please see, the inactivation curves where the data group in two clearly different types.

Another possible general arrangement between VSD and PD would happen if the channels were not in a domain-swapped configuration between subunits (protomers in the dimers). We also do not consider this as a possibility since, as in Shaker dimers in the present study, the probability of a different folding (non-domain-swapped assembling) would be similar to as in regular Shaker channels homotetramers. What supports this idea is that none of the residues that are part of the VSD//PD interface nor

VSD-PD linker (S4-S5 linker) have been replaced. In this regard it is important to note, however, that in at least one case of an ion channel (TRPV6) a residue in the VSD//PD interface when non-conservatively mutated, changes the channel from domain-swapping to non-domain-swapping channel (Singh et al., 2017. *Scientific Reports* PMID: 28878326).

B) Fig 2. Why do td-SHKN-ter and td-SHKC-ter have different inactivation properties? Apparently, the tandem constructs created asymmetry in the tetrameric channel. This asymmetry weakens any conclusion drawn from the experiments using these tandem dimers.

This is also an important point and therefore we address this point in this new version. The reviewer might be correct in saying that just introduction of W434F mutation in one of the protomers of the dimer may introduce asymmetries in the channels. We believe that this feature is most likely related to the production of heterodimers. Our data indeed show that when W434F mutation is in protomer 2 it appears less effective in speeding up inactivation than when is in the protomer 1. Although our conclusions are mostly based on the voltage dependence of the inactivation process we also discussed the differences in time constants. The possible reasons for the differences in the inactivation curves shown in Fig. 1h. were included in the discussion section in the new version of the manuscript (subsection “*The dimers make functional channels similar to homotetramers*” line 433 in the discussion section). In that section we discuss what are

Nevertheless, our data is clear in that, despite the differences shown in Fig. 1, we could still show how the inactivation process is controlled by the voltage sensor that is *near* the pore domain containing the W434F mutation (*vs* when it is *far*). These conclusions are based on a new set of experiments added to the manuscript (orange symbols in Figs. 3e-f). They show that regardless where W434F mutation is, in protomer 1 or 2, the inactivation curve is consistently split when the mutant VSD is *near* the pore domain containing W434F mutation. This is not true when a mutant VSD is positioned far from mutant PD. (green symbols in Figs. 3e-f).

C) The allosteric effects on the “far connection” need to be excluded when the “near connection” is considered.

This is another important topic to be addressed and we appreciate the question. Nonetheless, it was not in the scope of this work to move in that direction since several techniques would have to be attempted in order to exclude “far connection” allosteric effects, what we believe to be actually the canonical connection between VSD and PD from same subunits by the S4-S5 linker. It is in our plans, in a near future, to address this topic by using techniques that enables the “digestion” of the polypeptide chain at the level of the S4-S5 linker. We added a few words about in the discussion section with subtitle “*Evidence of non-canonical coupling for activation*”, line 489.

2. The manuscript compares properties of td-SHK_{near} and td-SHK_{far} against a number of other forms of the channel. The conclusions are derived from either the slope of the

comparison (e.g. Fig 4) or r^2 e.g. Fig 5, 6). This approach is novel and interesting. On the other hand, the interpretation of the results need to be more clearly developed for readers to follow.

In the new version of the manuscript, we eliminated the linear regressions we have used before. Instead, we decided to show how a change in the voltage dependence of the VSD (Q-V voltage dependence, V_0 and V_1) is reflected in the voltage dependence of both the activation ($V_{1/2}$) and the inactivation (V_{inact}) processes (Fig. 4). We plotted the shifts in the voltage dependences V_0 or V_1 against shifts in $V_{1/2}$ and V_{inact} , and observe the interdependence between them, especially when data come from *near* configured channels. Linear correlations could still be fit to the data, but the meaning of the outcomes is not clear, adding unnecessary complications.

These include:

A) The authors may want to explain the rationale of using either slope or r^2 , but not both, for interpretation of the results, and describe more clearly the physical significance behind the phenomena. What do the slope and r^2 indicate on the VSD-PD interactions?

We appreciate the comment, but, as explained in the previous paragraph, the new version does not include linear correlations analysis.

B) In Fig 4A results, it is not clear why td-SHK*far* has a shallower slope than td-SHK*wt-intact*. Please explain the results based on the comparison of all VSD-PD interactions. This result seems not to support the conclusion that the *near* interaction is important based on the slope.

Thanks for pointing this out. In the new version of the manuscript this matter appears both in Fig. 4e (W434F-Shaker dimers, *far* configured) as well as in Fig. 5d (wt-PD Shaker dimers, former td-SHK*wt-intact*). What is referred to as “shallower” slope in our interpretation is because a given VSD^{mut} (through the shift it causes in Shaker, for instance (please see Table 1 for all curves midpoints $V_{1/2}$)) seems to be more effective in inducing shifts in the activation curves in wt-PD Shaker dimers than to those curves in W434F-Shaker dimers *far* configured. Therefore we hypothesize that, because in W434F-Shaker dimers *far* configured the VSD//PD^{W434F} interfaces always contain a wild type VSD (always same V_0 and V_1 voltage dependent parameters), the voltage dependence parameters $V_{1/2}^*$ from several Shaker dimers activation (accounting for several different VSD^{mut}) are much less diverse. In this regard, a few words were added to the discussion section of the manuscript, starting in line 494.

C) Line 177. “we find that only V_1 had significant correlation”. Please define a value of r^2 for “significant correlation”.

Thanks for this comment but we are no longer using correlation coefficients, as mentioned above.

D) In Fig 4, 5, 7, V0 V1 of SHKW434F are compared to properties in td-SHK. The former has four subunits with W434F but the latter has only two. Is the Q-V (V0 and V1) the same in these two sets of channels?

We thank you for this comment. We do not show Q-V curves for these conductive inactivating Shaker dimers (former td-SHK), therefore this pairwise analysis is not possible. Since these channels are conductive and inactivating, the recordings of gating current from them become difficult. The parameters V0 and V1 are exclusive from non-conductive Shaker (former SHK_{W434F}), and V0* and V1* (former V0-dimer and V1-dimer, respectively) are from non-conductive Shaker dimers. Because we do not have Q-Vs from W434F-Shaker dimers (former td-SHK), we assumed that V0 and V1 from Shaker homotetramers containing a given VSD^{mut} are a good approximation of the voltage dependence of this particular VSD^{mut} when it is used in a W434F-Shaker dimer.

E) The authors seem to conclude that in Fig 8 the two components of Q-V of td-SHK_{allW434F} are derived from two different pairs of VSD. This assignment needs experimental justification.

We appreciate this comment because it prompted us to further analyze these results. Considering the Q-V curves from non-conductive Shaker dimers (former td-SHK_{allW434F}) now shown in Fig 5a are from gating currents from two populations of VSD in equal amounts, wild type and mutant, we fitted two independent two-state models equally weighted by a factor of 0.5. The fittings were all successful with low absolute sum of square residuals values. This argument, together with the two voltage dependences for each curve that are in agreement with a wild type and a mutant VSD in each curve, are in agreement with our claims that from these channels we can record gating charge movement from two populations of VSD at once. We included discussion about this matter in the subsection “*Cooperativity between subunits?*”, starting on line 501.

In addition, raw data for Figs 7 and 8 need to be shown.

The data of old figures 7 and 8 have been plotted in new Figures 3 and 5. New Figure 3 now has samples of raw traces used to plot inactivation and activation curves. Samples of raw gating current traces used to compute Q-V of Fig. 5 are now shown in Supplementary Figure S2.

3. In describing the Fig 3 results the authors state: “The broader range of voltages spanned by G-V and Inact-V curves from td-SHK_{near} channels resemble the voltage-dependences of the regular SHK bearing those respective mutations in the VSD.” What does “resemble” mean here?

This is another very important point to be revisited, thanks for it. Since we rewrote the manuscript, now this is stated as (starting line 253) “In the near configuration dimers the inactivation curves are consistently more dependent on changes in the voltage dependence of Q-V curves induced by mutations in the VSD (Fig. 4b) than in the dimers in far configuration (Figs. 4c)” and (line 301) “Shifts in voltage dependences of G-V curves from dimers were clearly larger in near

configured dimers than in far configured dimers as a consequence of the presence of a certain VSDmut.”

Since td-SHKnear only contains half of the “near connections” of SHK, is it expected that the results from the two channels should differ?

Correct, this is a very important point and we appreciate it was noted. The results are indeed different and this is now shown in Fig. 4. An example that clearly shows the referred difference is the comparison between values plotted in Fig 4a, V1/2 label and the midpoints of the plots showed in Fig. 4d (values can be found in Table 1 as V1/2 W434F-Shaker dimers *near* configured). The comparison to be made is essentially between the shift attributable to the VSD^{mut} (four times) in Shaker and the shift attributable to the same VSD^{mut} (two times) in W434F-Shaker dimers. On average, these percentual shifts are of 50±0.1% in wt-PD Shaker dimers, 61±0.1% in W434F-Shaker dimers *near* configuration and 24±0.1% in W434F-Shaker dimers *far* configuration (See graphs below).

Fig. R2. Shifts in the voltage dependence of conductance-voltage (G-V) curves from different Shaker dimers. *Left*, percentual shifts in the voltage-dependence of G-V curves from wt-PD, W434F-Shaker dimers, *near* and *far*, relative to the shifts induced by the same mutations, as indicated by colors and shapes, in Shaker. *Right*, average values as well as the individual data points (same as in Left panel) showing the average percentual shift per dimer type is less pronounced in W434F-Shaker dimers *far* configured.

4. The authors state that the S4-S5 interface is important for the inter-subunit “near” interaction, and show data and discussion of S412 (Fig 9) to support this statement. However, the data in Results focus on the interaction between VSD residues with W434F in the PD. Are these interactions considered important for the inter-subunit “near” interaction, and are they part of the S4-S5 interface in authors’ mind?

We acknowledge this is an important point and we thank you for noticing it. We believe yes, W434F is part of the non-canonical VSD-to-PD connection, but as part of a network that non-covalently connects the VSD to the PD. We hypothesize that the connection from S4 to the P-loop where W434F mutation is located includes the residues (Shaker numbers) 363, 366 or 369 in S4, 409 and 412 in S5, and 433 and 434 in the P-loop. (See in discussion starting on line 530).

The addition of the S412 data and discussion is confusing, and it is not a complete study to support the importance of S4-S5 interface in VSD-PD coupling. The evidence for the residue being important in the coupling is not strong and subjects to other interpretations.

Another interesting comment and we respectfully wish to argue that the residue S412 is an important, but only a putative element in the VSD//PD interface. Exactly because its mutation changes so much the voltage dependence, we believe it is part of the VSD-to-PD connection. Indeed, this residue equivalent has been naturally mutated and the appearance of the change in genomic sequences of human patients has been related to various diseases such as epilepsy, neurological seizures and arrhythmias. We carefully insist the S412 in Shaker as a prototype, may become an interesting tool for the study of the non-canonical connection between VSD and PD, with consequences for the activation and inactivation. This is the first step of a future work. (see in discussion starting on line 544).

5. The manuscript is hard to follow and some revisions may help readers like this reviewer. Some examples are:

A) Fig 3. Add a figure of structure showing all the residues 358, 381 and 434 (probably 412 as well).

Thanks for your suggestion. We included a new fig (Fig. 7) in the discussion section showing the relative positions of these residues.

B) Fig 4,5. A diagram showing different channels with mutations will be useful. The mutations need to be revealed and labeled in the existing panels.

We have added a few diagrams of the channels in Figs. 1 and 2 and we believe that those help to clarify this comment. We truly thank you for it.

C) Nomenclature V1/2(1), (2) in the text are different from those in Fig 8.

We thank for this point as well. We have improved the manuscript wording throughout as well as we were much more careful in matching the nomenclature as required.

D) Lines 310 and 311. Use either KCNQ1 (KCNQ2) or Kv7.1 (Kv7.2).

This nomenclature was cleared up in the new version of the discussion. Now these channels are mentioned starting on line 546.

6. The discussion in lines 310-316 on channels other than SHK is speculative and not supported by any analyses.

We acknowledge we were too assertive previously and now for the resubmission version we softened the language as one can see in the discussion section starting on line 544.

Minor concerns.

1. Fig 2, panel C mislabeled.

The manuscript was rewritten and we believe there is no mislabeling anymore.

2. Line 155. Check the reference.

In the new version this issue was fixed.

3. Line 421, 422: "Fig 6" should be "Fig 7".

In the new version this issue was fixed.

Reviewer #3 (Remarks to the Author):

This manuscript addresses the voltage-sensor pore coupling by using tandem dimers and electrophysiology recordings. The main finding of the paper is that the S4/S5 interface controls voltage-dependence of C-type inactivation and transduction of VSD voltage-dependence to the open pore probability.

The question is of broad interest and the approach used could be very interesting to the field but I find it difficult to verify that the claims are substantiated by the data due to quite severe communication problems that obscure the message:

1- The nomenclature of the different constructs and mutants is unclear. Please provide a table with the different constructs listed, the mutations present and make an attempt to make the names of the constructs informative and consistent.

We are grateful to you in raising this point and we have addressed the question in our new version of the manuscript. Specifically, a table has been added to the manuscript.

Indeed the nomenclature seems to be shifting along the paper. For example on line 192, V0, V1 and V1/2 is said to be calculated from SHK, Table 1 indicates that V0 and V1 are measured in SHKW434F and V1/2 in SHK while Fig 5.D indicates that V1/2 is evaluated in td-SHK. Sometimes VSDmut is used to indicate a construct, and sometimes to talk about the fact that the VSD is mutated within a construct.

We thank the reviewer for this comment. We renamed all different channels and parameters to what we now believe to be a much better way to do so. The new version has been written with that in mind.

2- The organization of the data is confusing: new experiments are introduced in the discussion section. Section titles do not always reflect the contents of the section, the first result section does not have a title which does not allow me to understand the main point made.

Thanks for pointing this out; we believe our new version of the manuscript has that problem solved. Now the version has subtitles in the results section and the manuscript has been reviewed by two colleagues who helped in making the presentation more coherent.

3- Purely descriptive statements about results are not discussed, either in the results section or in the discussion, as far as I can tell. As examples (but many others can be cited): l178: In addition, the V1 in the near case correlated better than the far case (Fig. 4B). End of paragraph. l198: The only strong correlation found was between parameters from td-SHKnear, where the tau slow correlated well with the V1/2 of the conductance activation. End of paragraph. Conversely, seemingly crucial aspects are neither mentioned nor discussed: for example, why are some meaningful correlations positive and others negative?

We understand this is a real difficulty in the previous manuscript therefore in our new version these problems have been addressed. It is important to note that we did not explore r^2 in the new version.

3- In several instances, the data presented in the text and tables is not compatible with the figures, as if new data points had been recorded but the numbers not updated consistently. The correlation coefficients in Figure 5 and Table 1 are different, I believe they should represent the same data. In figure 8, V0 and V1 are presented, the text talks about V1/2(1) and V1/2(2), which I believe are the same things and the r^2 presented on l.267 is different from the one in Fig.8. These are only examples, the manuscript should be thoroughly checked for consistency. Fig 4.B seem to contain two blue and two black data points at V \sim 80mV, it is difficult to understand why.

Since the manuscript was re-written without r^2 analysis because we decided it does not help the discussion, this problem was eliminated. Regarding the “double \sim 80 mV” problem that now is in Fig. 4h as a shift (\sim 25 mV), that happened because the V0 values from ILT (1st) and from 358W mutant VSD are very similar to each other as shown now in Table 1.

4- The variables used for different measurable quantities and model parameters sometimes change between the text, the caption and the graph axes (Fig. 8 for example).

We feel sorry for this mistake and appreciate your comment. We have solved this problem in the new version.

5- The language is also in several instances difficult to understand. As an example, I did not understand the following statements: I171: "These numbers show that in the near dimer case, the presence of two VSDmut influences the voltage-dependence of the channel conductance activation more than what would be expected for the abundance of VSDmut";

The manuscript was completely rewritten, therefore we believe this problem has been solved.

I182: "The plot of the first inactivation component's amplitude against V_0 , but not V_1 from VSDmut, gives a linear correlation with $r^2=0.68$ for the case of td-SHKnear and with $r^2=0.20$ for td-SHKfar (Fig. 5A).

We changed the analysis of the inactivation curves and now we have only one value for the voltage dependence of almost all curves (except the split ones in 361A and ILT). The r^2 analysis was also removed therefore we believe these problems were both solved.

I am willing to review a revised version if a serious attempt at clarifying and organizing the information is made because of the relevance and the importance of the topic. It might be a good idea for the authors to give it to a scientist which is not immediately from the field to proof-read it.

Thanks for this suggestion. We did follow your advice

Reviewer #4 (Remarks to the Author):

In Shaker channel, and most probably in many other voltage-gated channels, coupling between the voltage sensor domain VSD and the pore domain PD is, at least partly, realized by the linker between the two domains, S4-S5. The role of S4-S5 as a mechanical lever has been suggested in many studies. Another component of the coupling is made possible by the swapped domain arrangement of Shaker, which put the S4 voltage sensor of one subunit in direct apposition to the S5 of the adjacent subunit. In the present manuscript, Carvalho-de-Souza and Bezanilla use tandem dimers of Shaker channel to test if this non-covalent interaction between S4 of one subunit and S5 of adjacent subunit plays a role in the VSD/PD coupling.

The present manuscript provides new perspectives for the non-canonical mechanism of VS/PD coupling, but the results obtained here require additional experiments/analysis.

1. In a previous work published in JGP, the same group has described that a mutation in Shaker PD (W434F) drastically influences the effect of one mutation in the VSD (L361R), but without detailing the molecular mechanism. In particular it is not known if the observed functional interaction is due to the "far connection" between the VSD and PD of the same subunit, and/or the "near connection" of VSD/PD of adjacent subunits. Here, the use of dimer to discriminate the role the two connections seems relevant and elegant, but some observations raise some doubts about the conclusions.

In the td-SHKnear dimeric construct (Figure 1), PD mutation W434F is inserted in the first protomer. In the td-SHKfar construct, the same mutation is inserted in the second protomer. In the first protomer, the “C-terminus” of PD is linked to the VSD of the second protomer, whereas in the second protomer, the “C-terminus” of PD is free. This asymmetry is probably the cause of a major difference in the inactivation properties between the 2 controls (td-SHKN-ter and td-SHKC-ter), that present the W434F mutation in one of the PDs but no mutation in VSD (figure 2). This suggests that intrinsic stability of the PD is different between the 2 controls. Then it becomes impossible to conclude that the differences between the td-SHKnear and SHKfar activation/inactivation curves, observed when a mutation is introduced in VSD (figure 3) are related to the “far connection” or “near connection” between the VSD and PD. It may only be due the different intrinsic stability of the PD of the 2 constructs.

We thank you for this comment. The intrinsic stability of the PD in the dimers without mutations in their VSD (controls) cannot be considered different mainly because the activation (G-V) curves are practically identical. The difference between them, however, is in the inactivation process and we argue in this manuscript that part of the inactivation is independent on the open state (something different from classical C-type inactivation). Nevertheless, we are comparing the shifts in the inactivation curve compared to its related control, therefore we believe it is valid. Comments in this regard have been added to the discussion section, starting on line 433).

One way to overcome this first issue, would be to realize the same experiments on a different background: in both constructs the mutated PD would be in the second protomer (PD with the free C-ter). VSD mutation would be in VSD1 for td-SHKnear and in VSD2 for td-SHKfar. In that case, the two controls (no mutation in VSD) may be more similar than the ones presented in Figure 2.

This is another very interesting point that we thank you for the mentioning it. We actually performed the experiments you indicated by placing VSD mutation in the first and PD mutation in the second protomers, intending to have Shaker dimers in *near* configuration but with mutant PD C-terminal free. In Fig. 3 a-h, data show that no matter where VSD and PD are mutated in the dimer, the inactivation curve is split, very different from dimers intended to form *far* configured Shaker dimers, which was also made with a PD C-terminal free. It is interesting to note that in *near* configured dimers the first component of the inactivation curve is more pronounced when the mutant PD is in the first protomer (C-terminal linked to the VSD of the second protomer) as compared with the curve from channels with mutant PD in the second protomer (free C-terminal). This result is in agreement with the notion that the pore inactivation is possibly influenced by the mobility of the C-terminal of the same PD. (Please see discussion starting in line 433) Therefore, there are still differences between *near* configured dimers when PD is mutated in protomer 1 or in protomer 2 but qualitatively, meaning curve splitting feature, the data of both variants is very similar.

2. Alternatively, the observed influence of W434F on VSDmut may be due to the interaction between adjacent S1 and PD, which has been suggested by statistical

coupling analysis (Lee SY, Banerjee A, MacKinnon R. PLoS Biol. 2009;7(3):e47). The results presented here cannot exclude that the S1/PD interaction is a major component of the VSD/PD coupling. Similarly to the S4/S5 interaction, this interaction may participate to the differences between the td-SHK_{near} and SHK_{far} activation/inactivation curves.

Thanks for this note. Indeed the results by Lee et al. indicate that the interaction S1-PD is part of the VSD-to-PD coupling mechanism and we comment on that in the discussion of the new version (line 541) of our manuscript. Interestingly, according to this paper, the S1 from one subunit seems to be coupled to the PD from another subunit, this being an intersubunit interaction and therefore also part of the VSD//PD interface we are proposing here.

3. In the previous work in JGP, the 4 W434F mutations (far+near) in the tetramer induce a right shift in QV curve of L361R. The model would be strengthened if, in K-depleted solution, the 2 “near” W434F mutations also induce a right shift in QV curve of L361R.

We acknowledge the commentary that this experiment would be optimal, but it is impractical because “the channel open time decreases, the probability of a channel opening decreases, and the rate of inactivation increases” (J Gen Physiol. 1997 Jun 1; 109(6): 779–789). For these reasons an experiments to deplete K⁺ and to record gating currents in these Shaker dimers are impractical.

Also for 412V mutant, it would interesting to test the variation in coupling using tandem constructs in K-depleted conditions.

Thanks again, but we would have similar problems as in the previous answer.

4. A major part of the conclusions relies on correlation and r² comparisons, but the p-value of the correlation, which is provided by Prism, is never indicated.

We have eliminated the r² analysis from the new version because we think that the analysis was not robust, especially because we did not have many points in each graph for a linear regression.

Second, in line 181 it is stated that the first component of inactivation is correlated with VSD movement, partly based on r² of A1 with V0 (Figure 5A). However, the second component (1-A1) should give the same correlation.

This is an important point and therefore we appreciate it was mentioned. In the new version we changed the analysis of the inactivation curves. Now we only assumed that the inactivation curve was split when the residuals were too high (basically the absolute sum of the squared residuals were higher than 0.01 (See Fig. 4f,g)

Last, it would be useful if the author justify the use of AE1 and AE2 parameters.

For the new version of the manuscript we did not include this derived parameter AE.

Minor

5. Line 143: a figure would be useful for the comparison.

We changed the wording in this section, but we believe we answered your question with the Figs. 4a,b,d,h and j in the new version of the manuscript.

The manuscript contains many mistakes, some of them, render it difficult to read. Here are some.

1. Line 115 Equation 2, not 1
2. In figure 2, a panel is mislabeled. Figure 2C, not B
3. Line 150, "starts with negative voltages" is not clear
4. Lines 189-192, please explain in more details since data in Figure 5 and table 1 corresponds to different conditions. This is not clear in the present form.
5. Line 198 Figure 6C and D not 5C and D.
6. Line 263. Does $V_{1/2}(1)$ correspond to V_0 -dimer in figure 8?
7. Line 278 : Figure 4A not 3C
8. Line 345 : were not was
9. Line 349 : delta 6-46
10. Line 370 : "band" is lacking, kbp and not kDa. cf also line 378.
11. Line 372 : dephosphorylation
12. Line 373 : AP-dephosphorylated not digested
13. Line 382 : a promoter does not enhance transcription
14. Line 399: sentence is not clear. "Supplied" should be replaced by "decreased" and "uncompensated current" by "remaining capacitive current"
15. Lines 416-419 are probably inappropriate since another protocol follows in the next paragraph, with a different HP.
16. Lines 436-438: repetition of the same idea.
17. Line 474 averaged
18. Please remove all the 's

We thank you for your thorough reading of the manuscript. We have rewritten the paper and we also took many comments of two colleagues who read it. We made new figures and in doing so we have tried to minimize possible errors.

Reviewers' Comments:

Reviewer #1:

Remarks to the Author:

The authors have done an outstanding job of rewriting the manuscript to address the extensive reviewer comments and I think the work is now ready for publication. The story remains quite complex and it will not be easy for many to understand, but I think it will stimulate thinking and more experiments on the non-canonical coupling mechanism.

Reviewer #2:

Remarks to the Author:

This manuscript uses Shaker K⁺ channel tandem dimers to show that the voltage sensor domain and the pore domain of adjacent subunits may interact to pass functional coupling for inactivation and activation gating, which is a non-canonical VSD-PD coupling mechanism. While such a non-canonical VSD-PD coupling mechanism has been described in previous studies, the tandem dimers approach provides a structural and functional separation between the canonical and non-canonical mechanisms, which is novel. The authors also speculate that the residue S412 in S5 is a key residue, which may interact with residues in S4 for the non-canonical VSD-PD coupling mechanism. The data are of high quality and clearly presented. However, instead of speculating, the authors may want to do some additional experiments to demonstrate the importance of the residue S412 in the non-canonical VSD-PD coupling mechanism.

Reviewer #3:

Remarks to the Author:

The authors have made significant efforts to clarify the message presented in this paper and the present version is very much improved. The results are now mostly clearly presented and the conclusions are supported by the data. The most interesting aspect of the paper is the methodological one, since the conclusions according to which the near connection is important for VSD pore coupling via a non-canonical pathway are not novel, and have already been probed, notably by reference 9.

Altogether, the paper leaves as impression that while the methodology could be interesting to apply to other, less understood, channels, for the Shaker model channel, the novel insights are minimal. One of the strengths of the paper is probably to highlight that mutagenesis and other such techniques often generate non-straightforward insights because they can be much more invasive than initially thought: in particular the W434F mutants, combined to dimer constructs, highlight that a simple functional readout (inactivation) can have multiple structural and dynamical origins. The paper is, however, not written to highlight this.

More specific points:

In the introduction the authors present the questions as if they are the first to ask them: "1.65 " we speculated that allosteric PD-to-VSD interactions could also be mediated through a hypothetically functional VSD//PD interface". I would argue that this point is already relatively well established (refs 8,9).

The authors forgot to answer my point about the uncertainty related to the assembly of the dimers in a diagonal fashion, but provided a detailed answer to another reviewer without introducing changes in the paper. It think many readers will have this question, and a discussion section about this should be introduced in the paper.

An effort is still needed to clarify the contents of paragraphs p 14-16, and the description of Figure 5. I listed minor comments below that may help.

The last section of the results on S412 is in my view unnecessary, it does not make use of tandem dimers and probes a positions that has already been shown in ref. 9 to be important for the non-canonical coupling mechanism. The authors both write in the results that this residue “may be only one of several other possible candidates that affect the coupling” and in the discussion that S412 is “a key residue”. Because they have only probed this position, indeed, they cannot discriminate between these two possibilities.

Minor comments:

l.55: gating charge is not defined.

l. 193, 199: fig 2 should read fig 3.

Fig 3a-d: introduce clearer labels on the graphs, it is still a hassle to have to go through the details of the caption to understand what the panels are.

l. 274: define “better correlation”

l. 312 I may have misunderstood because I think there is a problem in the figure 5 caption, but I don't think the title describes what's in the paragraph below.

Fig 5a caption: Isn't this PDW434F instead of PDwt?

Fig 5b,c: Show same y-axis scale

l.355: The paragraph is lacking a conclusion and it is not clear how the data is linked to the title of this paragraph.

l.380: effect should be replaced by affect.

Reviewer #4:

Remarks to the Author:

I appreciate the effort in completely rewriting the manuscript. However most of my comments have not been addressed adequately. Even though the approach is novel and seems adequate to tackle the question, additional experiments/analyses are still missing.

1. In previous point 1., I was suggesting to study the mutations also in a second background: in both “near” and “far” constructs the mutated PD would be in the second protomer (PD with the free C-ter). Comparing the effects of the mutant in the two background will alleviate any bias due to the asymmetry caused by the dimer, asymmetry that has also been mentioned by Reviewer 2 in point 1B.

Authors have indeed studied one of the mutant in these two backgrounds but it is not sufficient, especially because ILT is a particular mutation, in a different region (S4 as opposed to S3-S4) and with a distinct effect (a major “split” in inactivation curve) as compared to the other 6 mutations.

2. In previous point 3., I was suggesting to test if in “near” rather than “far” dimers, introduction of W434F would induce a right shift in QV curve of L361R. This right shift has been observed by comparing the four L361R monomers also carrying W434F mutation to four L361R monomers not carrying the W434F mutation, in K-depleted condition (JGP 2018, same authors, 150:3007-321, figure 7B). So the required experiment does not seem impractical to me. I do not understand the argument : “[We have seen that as one or two W434F mutations are incorporated into a Shaker channel tetramer, the single channel conductance remains essentially the same but] the gating properties change: the channel open time decreases, the probability of a channel opening decreases, and the rate of inactivation increases” (JGP1997; 109: 779–789). For these reasons an experiments to deplete K⁺ and to record gating currents in these Shaker dimers are impractical.”. Measuring gating current in the 2 x W434F background, in K-depleted solutions seems possible (See also Olcese et al, JGP 1997 110:579-89. Figures 3 & 4).

Also, I still think doing the same experiments in the S412V background would be a robust argument toward the role of S412 in the “near” coupling. Figure 6 in its actual state does not

provide any robust argument in favor of S412 implication in "near" coupling.

3.As I said in my first report, a major part of the conclusions relies on correlations (previous point 4.). Now, there is no attempt at all to validate correlations, which goes one step backward as compared to the previous version.

Minor.

Moreover, all the mistakes concerning the methods section that I mentioned in the first reviewing (points 9 to 17) have not been corrected...

Reviewers' comments:

Reviewer #1 (Remarks to the Author):

The authors have done an outstanding job of rewriting the manuscript to address the extensive reviewer comments and I think the work is now ready for publication. The story remains quite complex and it will not be easy for many to understand, but I think it will stimulate thinking and more experiments on the non-canonical coupling mechanism.

We deeply appreciate your final comments and we also wish our work contributes to the field of voltage sensor to pore domain coupling.

Reviewer #2 (Remarks to the Author):

This manuscript uses Shaker K⁺ channel tandem dimers to show that the voltage sensor domain and the pore domain of adjacent subunits may interact to pass functional coupling for inactivation and activation gating, which is a non-canonical VSD-PD coupling mechanism. While such a non-canonical VSD-PD coupling mechanism has been described in previous studies, the tandem dimers approach provides a structural and functional separation between the canonical and non-canonical mechanisms, which is novel. The authors also speculate that the residue S412 in S5 is a key residue, which may interact with residues in S4 for the non-canonical VSD-PD coupling mechanism. The data are of high quality and clearly presented. However, instead of speculating, the authors may want to do some additional experiments to demonstrate the importance of the residue S412 in the non-canonical VSD-PD coupling mechanism.

We really appreciate your comments on the quality of our data and the strategy we used to acquire them. With regard to S412, we are presently working on new experiments but we feel that this is a new project, which is beyond the scope of the present paper.

Reviewer #3 (Remarks to the Author):

The authors have made significant efforts to clarify the message presented in this paper and the present version is very much improved. The results are now mostly clearly presented and the conclusions are supported by the data. The most interesting aspect of the paper is the methodological one, since the conclusions according to which the near connection is important for VSD pore coupling via a non-canonical pathway are not novel, and have already been probed, notably by reference 9.

Altogether, the paper leaves as impression that while the methodology could be interesting to apply to other, less understood, channels, for the Shaker model channel, the novel insights are minimal. One of the strengths of the paper is probably to highlight that mutagenesis and other such techniques often generate non-straightforward insights because they can be much more invasive than initially thought: in particular the W434F mutants, combined to dimer constructs, highlight that a simple functional readout (inactivation) can have multiple structural and dynamical origins. The paper is, however, not written to highlight this.

We thank you for your comments. Indeed other studies including the one by the authors from reference 9 stated the notion of a non-canonical coupling between voltage sensors and pore domain. Nevertheless, our data was presented as a short talk to the 61st Biophysical Society meeting, in 2017 (DOI: <https://doi.org/10.1016/j.bpj.2016.11.892>, reference mentioned in l.44 and l.69). In this past groundwork we set the basis of our present manuscript. The approach we have taken is different from the one used by the authors from reference 9. We studied the existence of a putative non-canonical coupling mechanism in a conducting channel, and as such, we were able to have information from the PD when we manipulate what we believe is the active interface between VSD and PD. It is also important to remark that in our study we managed to functionally and structurally separate the canonical from the non-canonical pathway, which was not done in reference 9. Therefore, this is the point that we respectfully submit as the novelty of our study.

More specific points:

In the introduction the authors present the questions as if they are the first to ask them: l.65 “ we speculated that allosteric PD-to-VSD interactions could also be mediated through a hypothetically functional VSD//PD interface”. I would argue that this point is already relatively well established (refs 8,9).

Thanks for this comment. Nevertheless, as stated above, we respectfully state that although other studies (references 8 and 9) have studied a putative non-canonical coupling mechanism in voltage dependent K⁺ channels gating, they do not present data, as we did, separating canonical from non-canonical mechanism, the reason to use the dimers strategy. However, we have changed the sentence starting l.69 by adding the words “*suggested by previous studies, including ours*”.

The authors forgot to answer my point about the uncertainty related to the assembly of the dimers in a diagonal fashion, but provided a detailed answer to another reviewer without introducing changes in the paper. It think many readers will have this question, and a discussion section about this should be introduced in the paper.

We apologize for this lapse of forgetting to answer to one of your comments. We included this issue in the discussion section, (l.480) and referred to it with a new supplementary figure (**Figure S3**).

An effort is still needed to clarify the contents of paragraphs p 14-16, and the description of Figure 5. I listed minor comments below that may help.

We are grateful for this call and we indeed changed the text of the paragraphs contained in the sub-section “*wt-PD and non-conductive Shaker dimers show independent VSD domains.*”, starting l.315. We believe these specific edits improved the understanding of the referred section.

The last section of the results on S412 is in my view unnecessary, it does not make use of tandem dimers and probes a positions that has already been shown in ref. 9 to be important for the non-canonical coupling mechanism. The authors both write in the results that this residue “may be only one of several other possible candidates that affect the coupling” and in the discussion that S412 is “a key residue”. Because they have only probed this position, indeed, they cannot discriminate between these two possibilities.

We appreciate this comment but we respectfully disagree with it. With that section, we wanted to show that the residue S412 is very important for the VSD-to-PD coupling, and we intentionally intended to demonstrate that with experiments in regular conductive Shaker, not the dimers, containing solely the mutation S412V. The referred study (reference 9) states that S412 together with V369 destabilize the active state of the voltage sensor of the channel. Our data shows that S412 stabilize the open state (a feature of the PD) since when we non-conservatively mutate it as S412V, the active state requires much more energy (more depolarization) to take place. In addition, in reference 9 studies the authors used mutation S412A in Shaker protein containing W434F mutation as well, therefore in non-conductive channels which only allows the readout from the VSD and not from the PD. Lastly, the analysis utilized by the authors from reference 9 takes a single voltage dependence parameter (V_{median}), thus disregarding intermediate states. This procedure has consequences for the analysis of the PD dynamics because it is the second step of the voltage sensor that is directly responsible to open the pore. Therefore, we believe that we cannot compare that study and ours, as presented here.

Minor comments:

I.55: gating charge is not defined.

Thanks for noticing that. We added a description for “gating charge” in I.56.

I. 193, 199: fig 2 should read fig 3.

Thanks! We fixed them, now in I.196 and I.202.

Fig 3a-d: introduce clearer labels on the graphs, it is still a hassle to have to go through the details of the caption to understand what the panels are.

We apologize for that and we made clearer labels to 3a-d. We also slightly changed Fig. 3 caption by adding the words “bearing VSD^{ILT}” to the first line of it.

I. 274: define “better correlation”

We replaced the term “better correlation” by “a mutual relationship or connection”, now in I. 277

I. 312 I may have misunderstood because I think there is a problem in the figure 5 caption, but I don't think the title describes what's in the paragraph below.

Thanks for this comment, we appreciate it. We changed our language in the main text and in figure 5 caption so that it agrees with the sentence, now starting in I. 316 for the main text and I.345 for the figure 5 caption..

Fig 5a caption: Isn't this PDW434F instead of PDwt?

Correct, we fixed that. Thanks!

Fig 5b,c: Show same y-axis scale

We really appreciate your comment on this matter, but when we scale y-axis of Fig 5b as in Fig 5c (-150 to +50) the data looks very crowded, with no enough resolution to appreciate the distribution of points along different voltages. Therefore, we have kept the Fig 5b y-axis scale as is.

I.355: The paragraph is lacking a conclusion and it is not clear how the data is linked to the title of this paragraph.

We appreciate your comment on this matter and we did include a conclusion sentence for the paragraph in question (please see I.368).

I.380: effect should be replaced by affect.

Thanks, we changed it and now it is in I.388.

Reviewer #4 (Remarks to the Author):

I appreciate the effort in completely rewriting the manuscript. However most of my comments have not been addressed adequately. Even though the approach is novel and seems adequate to tackle the question, additional experiments/analyses are still missing.

1. In previous point 1., I was suggesting to study the mutations also in a second background: in both “near” and “far” constructs the mutated PD would be in the second protomer (PD with the free C-ter). Comparing the effects of the mutant in the two background will alleviate any bias due to the asymmetry caused by the dimer, asymmetry that has also been mentioned by Reviewer 2 in point 1B. Authors have indeed studied one of the mutant in these two backgrounds but it is not sufficient, especially because ILT is a particular mutation, in a different region (S4 as opposed to S3-S4) and with a distinct effect (a major “split” in inactivation curve) as compared to the other 6 mutations.

We understand your concern. We established with the ILT triple mutations (a way to bias the voltage-dependence of the VSD) that the voltage-dependence of the inactivation process is split in two components. We demonstrate that in different dimers designs, no matter where VSD^{ILT} and PD^{W434F} is located, in protomer 1 (free N-terminus AND no free C-terminus) or in protomer 2 (no free N-terminus AND free C-terminus), the voltage-dependence of the inactivation curve is similarly split provided they are forming the same interface (*near* channels). With that we truly believe we showed convincing data for our claim that VSD controls at least part of the inactivation process by non-covalently “touching” a PD – this is the main novelty of our work. It is true that ideally the whole series of mutations we presented could be tested the same way, but as the data of those mutations are confirmatory of the ILT mutation conclusions we feel it is hard to justify the time that would be required to perform those experiments.

2. In previous point 3., I was suggesting to test if in “near” rather than “far” dimers, introduction of W434F would induce a right shift in QV curve of L361R. This right shift has been observed by comparing the four L361R monomers also carrying W434F mutation to four L361R monomers not carrying the W434F mutation, in K-depleted condition (JGP 2018, same authors, 150:3007-321, figure 7B). So the required experiment does not seem impractical to me. I do not understand the argument : “[We have seen that as one or two W434F mutations are incorporated into a Shaker channel tetramer, the single channel conductance remains essentially the same but] the gating properties change: the channel open time decreases, the probability of a channel opening decreases, and the rate of inactivation increases” (JGP1997; 109: 779–789). For these reasons an experiments to deplete K⁺ and to record gating currents in these Shaker dimers are impractical.”. Measuring gating current in the 2 x W434F background, in K-depleted solutions seems possible (See also Olcese et al, JGP 1997 110:579-89. Figures 3 & 4).

Perhaps we did not explain this correctly. The depletion experiments are done by using 0 K⁺ in both sides and 0 K⁺ in the microelectrode and by constantly pulsing so that there is outward current for each pulse. In fact, it is not possible to eliminate the internal K⁺ by just holding the membrane depolarized because slow inactivation prevents the outward flux of K⁺, even when internal perfusion is used. In the case of the 2X W434F, each pulse produces only a fast transient current, therefore it is very hard to deplete the internal K⁺ in a reasonable time before rundown, and this rundown is much faster in the perfused cut-open oocyte. In our experiments with Olcese, Latorre and Stefani we K-depleted the conducting clone but there was no need to deplete the 4X W434F because it was non-conducting.

Also, I still think doing the same experiments in the S412V background would be a robust argument toward the role of S412 in the “near” coupling. Figure 6 in its actual state does not provide any robust argument in favor of S412 implication in “near” coupling.

We understand your concerns about this part of our work and we are thankful for pointing this out. Our claim that S412 is key to the non-canonical coupling interface we propose here is based on two pieces of information: 1) A non-conservative mutation in this residue (S412V) shifts G-V curves to more depolarized potentials, it also shifts Q-V curves in the same direction (using non-conductive Shaker channels) and remarkably S412V mutation seems to uncouple G-V from Q-V curves (in other words in S412V more depolarization for more gating charge displacement is needed to open the channel) as compared to curves from wt channels (See table 2 for voltage-dependent parameters from Q-V and G-Vs). 2) In at least three other potassium channels (KV1.1, KV 7.1 and KV7.2), natural variants of these channels are associated with phenotypes that strongly suggest problems in the channels functionality, and are consistent with the premise that S412 participate in the coupling.

3. As I said in my first report, a major part of the conclusions relies on correlations (previous point 4.). Now, there is no attempt at all to validate correlations, which goes one step backward as compared to the previous version.

Again, thanks for your comment. It is true that parametric correlations would be ideal. However we decided to eliminate that kind of analysis presented in the first version of our manuscript for at least

two reasons. First, we realized that the correlations were not necessarily linear in many cases, which could have been solved by using a non-parametric test instead, that in turn would introduce more uncertainty to our conclusions from the analysis. Secondly and more importantly, we also realized we did not have enough points for a robust analysis (we only have 8 on those plots: points WT + 7 mutants).

Minor.

Moreover, all the mistakes concerning the methods section that I mentioned in the first reviewing (points 9 to 17) have not been corrected...

We feel deeply sorry for the mistake. The new version of our manuscript contains all changes from your previous revision suggestions.

We reply to points 9 to 17 below

9. Line 349 : delta 6-46

Fixed (now l.640).

10. Line 370 : “band” is lacking, kbp and not kDa. cf also line 378.

Fixed (now l.661 and l.669).

11. Line 372 : dephosphorylation

Fixed (now l.663).

12. Line 373 : AP-dephosphorylated not digested

We fixed that, thanks! Now in l.664.

13. Line 382 : a promoter does not enhance transcription

Thanks for noticing this mistake. We fixed that and now it can be found starting in l.675.

14. Line 399: sentence is not clear. “Supplied” should be replaced by “decreased” and “uncompensated current” by “remaining capacitive current”

Thanks, we fixed it (now l.692 and l.693).

15. Lines 416-419 are probably inappropriate since another protocol follows in the next paragraph, with a different HP.

Correct, we fixed it. Now it is in l.711)

16. Lines 436-438: repetition of the same idea.

Thanks for this comment – it was now fixed.

17. Line 474 averaged

Fixed, now in l.768.

18. Please remove all the ‘s

Fixed.

Reviewers' Comments:

Reviewer #4:

Remarks to the Author:

Previous points 1 and 3.

I am still convinced that conclusions of the article are weakened by :

- previous point 3: a lack of correlations between $V_{0/1}$ shifts and V_{inact} shift,
- previous point 1: absence of test of the S3-S4 mutants in both background (PDW434F mutation in protomer 1 or 2, like experiments on ILT mutations, in Figure 3). ILT is a particular mutation, in a different region (S4) and with a distinct effect (a major "split" in inactivation curve) as compared to the other 6 mutations.

Thus, I think that studying at least one of the 6 S3-S4 mutants in both background is necessary to support the conclusion of the article.

Regarding previous point 2, I now understand the point of the authors, thanks to the detailed response:

" The depletion experiments are done by using 0 K⁺ in both sides and 0 K⁺ in the microelectrode and by constantly pulsing so that there is outward current for each pulse. In fact, it is not possible to eliminate the internal K⁺ by just holding the membrane depolarized because slow inactivation prevents the outward flux of K⁺, even when internal perfusion is used. In the case of the 2X W434F, each pulse produces only a fast transient current, therefore it is very hard to deplete the internal K⁺ in a reasonable time before rundown, and this rundown is much faster in the perfused cut-open oocyte."

Given the broad audience of the journal, such information should be given to the reader.

- Minor : OK.

REVIEWERS' COMMENTS:

Reviewer #4 (Remarks to the Author):

Previous points 1 and 3.

I am still convinced that conclusions of the article are weakened by :

- previous point 3: a lack of correlations between V0/1 shifts and V_{inact} shift,

We respectfully repeat our argument that the numbers of points, precisely eight, is not sufficient for such analysis.

- previous point 1: absence of test of the S3-S4 mutants in both background (PDW434F mutation in protomer 1 or 2, like experiments on ILT mutations, in Figure 3). ILT is a particular mutation, in a different region (S4) and with a distinct effect (a major “split” in inactivation curve) as compared to the other 6 mutations.

Thus, I think that studying at least one of the 6 S3-S4 mutants in both background is necessary to support the conclusion of the article.

We understand the concern of this reviewer. We have already answered this comment before.

However, we would like to add respectfully that to study the other six mutations, or even one of them, in near configuration with PDW434F in protomer 2 is unnecessary. It is known that S4 movements are not supposed to be affected by N- or C- termini from that domain's (or subunit) freedom to move. The literature does not report anything like that in multi-domains channels such as voltage dependent Na⁺ or Ca²⁺ channels. Also our data clearly show that the same S4 (wild-type S4) behaves exactly the same way with regard to voltage dependence in dimerized channels, where one VSD is in a different position relative to the other VSD in comparison to N- and C- termini. This tells us any of the six VSDmut would behave the same in protomer 1 or protomer 2. Therefore the expected difference between the two different near configuration dimers would be a slight change in the voltage dependence of the inactivation process, that is about 5 mV with a more negative V_{inact} when W434F is in protomer 1 (Please see Fig. 1c-d, h and Supplementary Table 1). In other words we expect no changes in voltage-dependence of inactivation process in near configured channels due to the same VSDmut in the VSD//PD interface, but in protomer 1 ($VSD^{mut}PD^{wt}$ - $VSD^{wt}PD^{W434F}$) compared to in protomer 2 ($VSD^{wt}PD^{W434F}$ - $VSD^{mut}PD^{wt}$). The change in voltage dependence of the inactivation in this comparison is solely due to the position of the W434F mutation, as shown for dimers with only VSD^{wt} (Fig. 1h) and when we tested for this hypothesis using VSD^{ILT} (Fig. 3).

Regarding previous point 2, I now understand the point of the authors, thanks to the detailed response:

“ The depletion experiments are done by using 0 K⁺ in both sides and 0 K⁺ in the microelectrode and by constantly pulsing so that there is outward current for each pulse. In fact, it is not possible to eliminate the internal K⁺ by just holding the membrane depolarized because slow inactivation prevents the outward flux of K⁺, even when internal perfusion is used. In the case of the 2X W434F, each pulse produces only a fast transient current, therefore it is very hard to deplete the internal K⁺ in a reasonable

time before rundown, and this rundown is much faster in the perfused cut-open oocyte.”

Given the broad audience of the journal, such information should be given to the reader.

Thanks for the positive feedback on this point. This paragraph was included in the Supplementary Information as Supplementary Note 1 and it is referred to it in the main text.

- Minor : OK.